# ENABLING EFFICIENT EQUIVARIANT OPERATIONS IN THE FOURIER BASIS VIA GAUNT TENSOR PRODUCTS

**Shengjie Luo**[1][†][*]        **Tianlang Chen**[2][*]        **Aditi S. Krishnapriyan**[2]
[1]Peking University        [2]University of California, Berkeley        [†]Project lead

## ABSTRACT

Developing equivariant neural networks for the E(3) group plays an important role in modeling 3D data across real-world applications. Enforcing this equivariance primarily involves the tensor products of irreducible representations (irreps). However, the computational complexity of such operations increases significantly as higher-order tensors are used. In this work, we propose a systematic approach to substantially accelerate the computation of the tensor products of irreps. We mathematically connect the commonly used Clebsch-Gordan coefficients to the Gaunt coefficients, which are integrals of products of three spherical harmonics. Through Gaunt coefficients, the tensor product of irreps becomes equivalent to the multiplication between spherical functions represented by spherical harmonics. This perspective further allows us to change the basis for the equivariant operations from spherical harmonics to a 2D Fourier basis. Consequently, the multiplication between spherical functions represented by a 2D Fourier basis can be efficiently computed via the convolution theorem and Fast Fourier Transforms. This transformation reduces the complexity of full tensor products of irreps from $\mathcal{O}(L^6)$ to $\mathcal{O}(L^3)$, where $L$ is the max degree of irreps. Leveraging this approach, we introduce the Gaunt Tensor Product, which serves as a new method to construct efficient equivariant operations across different model architectures. Our experiments on the Open Catalyst Project and 3BPA datasets demonstrate both the increased efficiency and improved performance of our approach.

## 1 INTRODUCTION

The imposition of physical priors, corresponding to specific constraints, has played an important role in deep learning (Jumper et al., 2021). To address the curse of dimensionality, prior works exploited the geometry inherent in data, such as pixel grids in images or atomic positions in molecules, along with their associated symmetries. Among the constraints introduced by these physical priors, equivariance has emerged as an important concept in deep learning (Bronstein et al., 2021).

A function mapping with the equivariance constraint transforms the output predictably in response to transformations on the input. One typical class of such transformations includes translations, rotations, and reflections in the Euclidean space, collectively termed the Euclidean group (Rotman, 1995). Numerous fundamental real-world problems exhibit symmetries to the Euclidean group, including molecular modeling (Atz et al., 2021), protein biology (Jumper et al., 2021), and 3D vision (Weiler et al., 2018). Many works have developed a rich family of equivariant operations for the Euclidean group (Han et al., 2022), demonstrating promise in challenging prediction tasks through both improved data efficiency and generalization.

Among the various approaches for enforcing equivariant operations (Cohen & Welling, 2016; 2017; Coors et al., 2018; Finzi et al., 2020; Satorras et al., 2021, see Appendix B for more advances), employing tensor products of irreducible representations (irreps) has recently emerged as the dominant choice for the Euclidean group (Thomas et al., 2018; Fuchs et al., 2020; Brandstetter et al., 2022; Batzner et al., 2022; Liao & Smidt, 2023; Musaelian et al., 2023). Rooted in the mathematical foundation of representation theory, such operations allow for the precise modeling of equivariant mappings between any $2l + 1$-dimensional irreps space via Clebsch-Gordan coefficients (Wigner, 2012), and are proven to have universal approximation capability (Dym & Maron, 2021). These operations have driven state-of-the-art performance across a variety of real-world applications (Ramakrishnan et al., 2014; Chmiela et al., 2017; Chanussot et al., 2021), demonstrating strong empirical results.

---

[*]Equal contribution. Correspondence to Tianlang Chen <tlchen@pku.edu.cn>, Shengjie Luo <luosj@stu.pku.edu.cn> and Aditi S. Krishnapriyan <aditik1@berkeley.edu>.

The computational complexity inherent to computing the tensor products of irreps is a well-acknowledged hurdle in achieving better efficiency, as well as greater accuracy (see comments in Fuchs et al., 2020; Schütt et al., 2021; Satorras et al., 2021; Frank et al., 2022; Batatia et al., 2022b; Brandstetter et al., 2022; Musaelian et al., 2023; Passaro & Zitnick, 2023; Liao et al., 2024; Simeon & Fabritiis, 2023; Wang & Chodera, 2023; Han et al., 2022; Zhang et al., 2023b). Recent advancements suggest that the empirical performance of state-of-the-art models based on such operations can be consistently boosted by using higher degrees of irreps (Batzner et al., 2022; Batatia et al., 2022b; Brandstetter et al., 2022; Passaro & Zitnick, 2023; Liao et al., 2024; Musaelian et al., 2023). However, the full tensor product of irreps up to degree $L$ have an $\mathcal{O}(L^6)$ complexity, which significantly limits $L$ to 2 or 3 in practice, hindering efficient inference on large systems. Given this, there is a strong need to improve the efficiency of the tensor products of irreps.

In this work, we introduce a systematic methodology to accelerate the computation of the tensor products of irreps. Our key innovation is established on the observation that the Clebsch-Gordan coefficients are mathematically related to the integral of products of three spherical harmonics, known as the Gaunt coefficients (Wigner, 2012). We use this to show that the tensor product of irreps via the Gaunt coefficients is equivalent to multiplication between spherical functions represented by linear combinations of spherical harmonic bases. This understanding motivates us to employ the 2D Fourier basis to equivalently express the same functions via a change of basis. Multiplications between functions represented by 2D Fourier bases can be significantly accelerated through the convolution theorem and Fast Fourier Transforms (FFT). Using these insights, the computational cost of full tensor products of irreps up to degree $L$ can be substantially reduced to $\mathcal{O}(L^3)$.

Building upon this approach, we introduce the Gaunt Tensor Product, which serves as a new method for efficient equivariant operations across various model designs. We provide a comprehensive study on major operation classes that are widely used in equivariant models for the Euclidean group, demonstrating the generality of our method and how it can be used to design efficient operations:

- *Equivariant Feature Interactions*. Operations in this category calculate tensor products between two equivariant features, enabling the interaction between irreps from different dimensional spaces and objects. These operations have been widely used in recent advances (Unke et al., 2021; Frank et al., 2022; Yu et al., 2023b), e.g., interacting atomic features and predicting a Hamiltonian matrix. Employing our approach in this scenario is straightforward.

- *Equivariant Convolutions*. Tensor products in this category are computed between equivariant features and spherical harmonic filters, which are the building blocks for equivariant message passing (Thomas et al., 2018; Fuchs et al., 2020; Batzner et al., 2022; Brandstetter et al., 2022). In this case, we demonstrate that additional speed-up can be achieved via the insight from Passaro & Zitnick (2023), which introduces sparsity to the filters via a rotation. We use this to further accelerate the conversion from spherical harmonics to 2D Fourier bases.

- *Equivariant Many-Body Interactions*. Modeling many-body interaction effects has been shown to be important for advancing machine learning force fields (Drautz, 2019; Dusson et al., 2022; Batatia et al., 2022a;b; Nigam et al., 2022). Capturing these interactions requires performing multiple tensor products among equivariant features. Our connection between Clebsch-Gordan coefficients and Gaunt coefficients is naturally generalized to this setting, and we develop a divide-and-conquer approach to parallelize the computation of multiple 2D convolutions, demonstrating better efficiency compared to original implementations.

We conduct extensive experiments to verify the efficiency and generality of our approach. For each of the aforementioned operation classes, we provide efficiency comparisons between our Gaunt Tensor Product and implementations via the e3nn library (Geiger & Smidt, 2022). We also apply our Gaunt Tensor Product to state-of-the-art models on benchmarks across task types and scales, e.g., force field modeling on OC20 (Chanussot et al., 2021) and 3BPA (Kovács et al., 2021) benchmarks. All results show both the better efficiency and improved performance of our approach.

## 2 BACKGROUND

We review the necessary background knowledge for our method, including concepts of the Euclidean group and equivariance, irreducible representations (irreps), spherical harmonics, and the tensor product of irreps. We include all the details of the mathematics used in this work in Appendix A.

Let $S$ denote a set of objects located in the three-dimensional Euclidean space. We use $\mathbf{X} \in \mathbb{R}^{n \times d}$ to denote the objects with features, where $n$ is the number of objects, and $d$ is the feature dimension. Given object $i$, let $\mathbf{r}_i \in \mathbb{R}^3$ denote its Cartesian coordinate. We define $S = (\mathbf{X}, R)$, where $R = \{\mathbf{r}_1, ..., \mathbf{r}_n\}$. These data structures are inherent in a variety of real-world systems, such as molecular systems and point clouds. Problems defined on $S$ generally exhibit symmetries to transformations on the objects. For instance, given a molecular system, if a rotation is applied to each atom position $\mathbf{r}_i$, the direction of the force on each atom will also rotate. Mathematically, such transformations and symmetries are directly related to the concepts of groups and equivariance that we will introduce below.

**Euclidean group and Equivariance.** Formally, a *group* $G$ is a non-empty set, together with a binary operation $\circ : G \times G \to G$ satisfying group axioms that describe how the group elements interact with each other. The class of transformations including translations, rotations, and reflections in the three-dimensional Euclidean space is of great interest, as it describes many real-world scenarios. These transformations form a group structure known as the Euclidean group, and denoted as $E(3)$.

To describe how the group elements act on vector spaces, *group representations* are used. Formally, the representation of a group $G$ on a vector space $V$ is a mapping $\rho^V : G \to GL(V)$, where $GL(V)$ is the set of invertible matrices on $V$. For example, representations of the rotation group $SO(3)$ on $\mathbb{R}^3$ can be parameterized by orthogonal transformation matrices $\mathbf{R} \in \mathbb{R}^{3 \times 3}, \det(\mathbf{R}) = 1$, which are applied to $\mathbf{x} \in \mathbb{R}^3$ via matrix multiplication $\mathbf{R}\mathbf{x}$.

We now introduce the concept of *Equivariance*. Let $\phi : \mathcal{X} \to \mathcal{Y}$ denote a function mapping between vector spaces. Given a group $G$, let $\rho^{\mathcal{X}}$ and $\rho^{\mathcal{Y}}$ denote its representations on $\mathcal{X}$ and $\mathcal{Y}$ respectively. A function $\phi : \mathcal{X} \to \mathcal{Y}$ is said to be equivariant if it satisfies the following condition $\rho^{\mathcal{Y}}(g)[\phi(x)] = \phi(\rho^{\mathcal{X}}(g)[x])$, for all $g \in G, x \in \mathcal{X}$. When $\rho^{\mathcal{Y}} = \mathcal{I}^{\mathcal{Y}}$ (identity transformation) for $G$, it is a special case of equivariance, known as *Invariance*.

The equivariance constraint is a commonly exploited physical prior in machine learning models. For tasks defined on the data structure $S$, it is typically important to ensure operations maintain $E(3)$-equivariance. The $E(3)$ group is a semi-direct product of the group of translations $\mathbb{R}^3$ with the group of orthogonal transformations $O(3)$, i.e., $E(3) = \mathbb{R}^3 \rtimes O(3)$. In general, achieving translation equivariance is straightforward through convolution-type operations. Therefore, we primarily focus on the orthogonal group $O(3)$, which consists of rotations and reflections.

**Irreducible representations.** We use $\mathbf{D}(g)$ to denote the representation of group element $g$ in the matrix form. For vector spaces of dimension $2l + 1, l \in \mathbb{N}$, there exists a collection of matrix representations indexed with $l$ for each $g \in SO(3)$, which are called *Wigner-D matrices*. They are denoted as $\mathbf{D}^{(l)}(g)$. Each Wigner-D matrix $\mathbf{D}^{(l)}(g)$ is the *irreducible representation* of $SO(3)$ on a subspace with dimension $2l + 1$. Any matrix representation $\mathbf{D}(g)$ of $SO(3)$ on vector space $V$ can be reduced to an equivalent block diagonal matrix representation with Wigner-D matrices along the

diagonal: $\mathbf{D}(g) = \mathbf{Q}^{-1}(\mathbf{D}^{(l_1)}(g) \oplus \mathbf{D}^{(l_2)}(g) \oplus ...)\mathbf{Q} = \mathbf{Q}^{-1} \begin{pmatrix} \mathbf{D}^{(l_1)}(g) & & \\ & \mathbf{D}^{(l_2)}(g) & \\ & & ... \end{pmatrix} \mathbf{Q}$. Here,

$\mathbf{Q}$ is the change of basis that makes $\mathbf{D}(g)$ and the block diagonal matrix representation equivalent.

This equivalence provides an alternative view that the representations that act on the vector space $V$ are made up of each Wigner-D matrix $\mathbf{D}^{(l_i)}$, which only acts on a $2l_i + 1$-dimension subspace $V_{l_i}$ of $V$. Then, the vector space $V$ can be factorized into $V_{l_1} \oplus V_{l_2} \oplus ...$ We call the space $V_l$ a *type-l irrep space*, and $\mathbf{x}^{(l)} \in V_l$ a *type-l irrep*. We have already introduced the Wigner-D matrix in the type-1 space, i.e., $\mathbf{D}^{(1)} = \mathbf{R} \in \mathbb{R}^{3 \times 3}, \det(\mathbf{R}) = 1$. The Wigner-D matrix representations can be naturally extended to the $O(3)$ group by simply including the reflections via a direct product.

**Spherical harmonics.** Spherical harmonics are used to map from $\mathbb{R}^3$ to any type-$l$ irrep space. We use $Y_m^{(l)} : S^2 \to \mathbb{R}, -l \le m \le l, l \in \mathbb{N}$ to denote the spherical harmonics, where $l$ and $m$ denote their degrees and orders respectively. For any $\mathbf{x} \in \mathbb{R}^3$, $Y^{(l)}(\frac{\mathbf{x}}{||\mathbf{x}||}) = [Y_{-l}^{(l)}(\frac{\mathbf{x}}{||\mathbf{x}||}), ..., Y_l^{(l)}(\frac{\mathbf{x}}{||\mathbf{x}||})]^\top$ is a type-$l$ irrep. This mapping is equivariant to the $O(3)$ group, i.e., given $\tilde{\mathbf{r}} \in S^2, g \in SO(3)$ and its representation on $\mathbb{R}^3$ as rotation matrix $\mathbf{R}$, we have $Y^{(l)}(\mathbf{R}\tilde{\mathbf{r}}) = \mathbf{D}^{(l)}(g)Y^{(l)}(\tilde{\mathbf{r}})$ and $Y^{(l)}(-\tilde{\mathbf{r}}) = (-1)^l Y^{(l)}(\tilde{\mathbf{r}})$.

**Tensor products of irreducible representations.** Given $\mathbf{x}^{(l_1)}, \mathbf{x}^{(l_2)}$ in type-$l_1$ and type-$l_2$ irrep space respectively, their tensor product yields $\mathbf{x}^{(l_1)} \otimes \mathbf{x}^{(l_2)}$, which lies in the $V_{l_1} \otimes V_{l_2}$ space with

$(2l_1 + 1) \times (2l_2 + 1)$ dimension. This operation preserves equivariance to the $O(3)$ group, i.e., $(\mathbf{D}^{(l_1)}(g) \otimes \mathbf{D}^{(l_2)}(g))(\mathbf{x}^{(l_1)} \otimes \mathbf{x}^{(l_2)}) = (\mathbf{D}^{(l_1)}(g)\mathbf{x}^{(l_1)}) \otimes (\mathbf{D}^{(l_2)}(g)\mathbf{x}^{(l_2)})$ for all $g \in O(3)$. As previously stated, the tensor product of irreducible representations $\mathbf{D}^{(l_1)}(g) \otimes \mathbf{D}^{(l_2)}(g)$ can be further decomposed into irreducible pieces: $\mathbf{D}^{(l_1)}(g) \otimes \mathbf{D}^{(l_2)}(g) = \oplus_{l=|l_1-l_2|}^{l_1+l_2} \mathbf{D}^{(l)}(g)$. The resulting irrep $\mathbf{x}^{(l_1)} \otimes \mathbf{x}^{(l_2)}$ of the tensor product can be organized via a change of basis into parts that are in different type-$l$ irrep spaces, i.e., $\mathbf{x}^{(l_1)} \otimes \mathbf{x}^{(l_2)} \in V = V_{|l_1-l_2|} \oplus ... \oplus V_{l_1+l_2}$.

The Clebsch-Gordan tensor product provides an explicit way to calculate the resulting type-$l$ irrep after the direct sum decomposition. Given $\mathbf{x}^{(l_1)} \in \mathbb{R}^{2l_1+1}$ and $\mathbf{x}^{(l_2)} \in \mathbb{R}^{2l_2+1}$, the $m$-th element of the resulting type-$l$ irrep of the tensor product between $\mathbf{x}^{(l_1)}$ and $\mathbf{x}^{(l_2)}$ is given by:

$$(\mathbf{x}^{(l_1)} \otimes_{cg} \mathbf{x}^{(l_2)})_m^{(l)} = \sum_{m_1=-l_1}^{l_1} \sum_{m_2=-l_2}^{l_2} C_{(l_1,m_1)(l_2,m_2)}^{(l,m)} x_{m_1}^{(l_1)} x_{m_2}^{(l_2)}, \tag{1}$$

where $C_{(l_1,m_1)(l_2,m_2)}^{(l,m)}$ are the Clebsch-Gordan coefficients (Wigner, 2012). We use $\tilde{\mathbf{x}}^{(L)} = [\mathbf{x}^{(0)}; ...; \mathbf{x}^{(L)}] \in \mathbb{R}^{(L+1)^2}$ to denote features containing irreps of up to degree $L$ in practice, and the full tensor product is given by $(\tilde{\mathbf{x}}^{(L_1)} \otimes_{\tilde{cg}} \tilde{\mathbf{x}}^{(L_2)})_m^{(l)} = \sum_{l_1=0}^{L_1} \sum_{l_2=0}^{L_2} (\mathbf{x}^{(l_1)} \otimes_{cg} \mathbf{x}^{(l_2)})_m^{(l)}$. Retaining all output irreps of the Clebsch-Gordan tensor product requires $\mathcal{O}(L^3)$ 3D matrix multiplications, i.e., an $\mathcal{O}(L^6)$ cost, which becomes computationally unfeasible when scaling irreps to higher $L$ degrees.

## 3 GAUNT TENSOR PRODUCT

We now introduce our Gaunt Tensor Product, which serves as a new method for efficient $E(3)$-equivariant operations. First, we propose a new perspective on the tensor products of irreps (Section 3.1), which connects this operation to multiplication between functions defined on the sphere via Gaunt coefficients (Wigner, 2012). Our perspective inspires the development of an efficient approach to accelerate the tensor product computation by using the convolution theorem and Fast Fourier Transforms (Section 3.2), which achieves substantial speed-ups compared to prior implementations. Finally, we provide a comprehensive study on major equivariant operation classes in the literature, and show how to design efficient counterparts via our Gaunt Tensor Product (Section 3.3), which opens new opportunities for pushing the frontier of $E(3)$-equivariant models.

### 3.1 A NEW PERSPECTIVE ON TENSOR PRODUCTS OF IRREPS VIA GAUNT COEFFICIENTS

As shown in Eqn. (1), the essence of the tensor product operation is in the Clebsch-Gordan coefficients, which are used to perform the explicit direct-sum decomposition. The CG coefficients commonly appear in quantum mechanics, and serve as the expansion coefficients of total angular momentum eigenstates in an uncoupled tensor product basis. Interestingly, spherical harmonics actually correspond to the eigenstates of angular momentum, indicating that there exist underlying relations between these two concepts. In physics, the Wigner-Eckart theorem in quantum mechanics reveals these relationships in an elegant manner (refer to Appendix A.7 for the details of these physical concepts):

**Theorem 3.1** (Wigner-Eckart theorem for Spherical Tensor Operators, from Jeevanjee (2011))**.** *Let $j, l, J \in \mathbb{N}_{\geq 0}$ and let $\boldsymbol{T}^{(j)}$ be a spherical tensor operator of rank $j$. Then there is a unique complex number, the reduced matrix element $\lambda \in \mathbb{C}$ (often written $\langle J \| \boldsymbol{T}^{(j)} \| l \rangle \in \mathbb{C}$), that completely determines any of the $(2J+1) \times (2j+1) \times (2l+1)$ matrix elements $\langle J, M | T_m^{(j)} | l, n \rangle$:*

$$\langle J, M | T_m^{(j)} | l, n \rangle = \lambda \cdot C_{(l,n)(j,m)}^{(J,M)}. \tag{2}$$

Here, $|l, n\rangle$, $\langle J, M|$ denote eigenstates for angular momentum operators with $|n| \leq l$, $|M| \leq J$, and relate to spherical harmonics via $\langle \theta, \psi | l, n \rangle = Y_n^{(l)}(\theta, \psi)$, with spherical coordinate $\theta \in [0, \pi], \psi \in [0, 2\pi)$. The Wigner-Eckart theorem describes how the matrix elements $(\langle J, M | T_m^{(j)} | l, n \rangle, |m| \leq j)$ of the spherical tensor operator $(\boldsymbol{T}^{(j)})$ in the basis of angular momentum eigenstates are decomposed into the Clebsch-Gordan coefficients and the reduced representation $(\langle J \| \boldsymbol{T}^{(j)} \| l \rangle)$. If we instantiate $\boldsymbol{T}^{(j)}$ as the spherical harmonics $Y^{(j)}$ and apply the Wigner-Eckart theorem, we obtain the following relation $\langle J, M | Y_m^{(j)} | l, n \rangle = \langle J \| Y^{(j)} \| l \rangle \cdot C_{(l,n)(j,m)}^{(J,M)}$, which can be rewritten as,

$$\int_0^{2\pi} \int_0^{\pi} Y_{m_1}^{(l_1)}(\theta, \psi) Y_{m_2}^{(l_2)}(\theta, \psi) Y_m^{(l)}(\theta, \psi) \sin\theta d\theta d\psi = \tilde{C}_{(l_1,l_2)}^{(l)} C_{(l_1,m_1)(l_2,m_2)}^{(l,m)}, \tag{3}$$

where $\tilde{C}^{(l)}_{(l_1,l_2)}$ is a real constant only determined by $l_1, l_2$ and $l$. From Eqn. (3), the Clebsch-Gordan coefficients are mathematically related to the integrals of products of three spherical harmonics, which are known as the Gaunt coefficients (Wigner, 2012). We denote them as $G^{(l,m)}_{(l_1,m_1)(l_2,m_2)}$.

If we use the Gaunt coefficients in the tensor product operation instead, we have the following result on the tensor product of irreps:

$$
\begin{aligned}
(\tilde{\mathbf{x}}^{(L_1)} \otimes_{\tilde{Gaunt}} \tilde{\mathbf{x}}^{(L_2)})^{(l)}_m &= \sum_{l_1=0}^{L_1} \sum_{l_2=0}^{L_2} (\mathbf{x}^{(l_1)} \otimes_{Gaunt} \mathbf{x}^{(l_2)})^{(l)}_m \\
&= \sum_{l_1=0}^{L_1} \sum_{l_2=0}^{L_2} \sum_{m_1=-l_1}^{l_1} \sum_{m_2=-l_2}^{l_2} G^{(l,m)}_{(l_1,m_1)(l_2,m_2)} x^{(l_1)}_{m_1} x^{(l_2)}_{m_2}, \\
&= \sum_{l_1=0}^{L_1} \sum_{l_2=0}^{L_2} \sum_{m_1=-l_1}^{l_1} \sum_{m_2=-l_2}^{l_2} x^{(l_1)}_{m_1} x^{(l_2)}_{m_2} \int_0^{2\pi} \int_0^{\pi} Y^{(l_1)}_{m_1}(\theta,\psi) Y^{(l_2)}_{m_2}(\theta,\psi) Y^{(l)}_m(\theta,\psi) \sin\theta \mathrm{d}\theta \mathrm{d}\psi, \\
&= \int_0^{2\pi} \int_0^{\pi} \left( \sum_{l_1=0}^{L_1} \sum_{m_1=-l_1}^{l_1} x^{(l_1)}_{m_1} Y^{(l_1)}_{m_1}(\theta,\psi) \right) \left( \sum_{l_2=0}^{L_2} \sum_{m_2=-l_2}^{l_2} x^{(l_2)}_{m_2} Y^{(l_2)}_{m_2}(\theta,\psi) \right) Y^{(l)}_m(\theta,\psi) \sin\theta \mathrm{d}\theta \mathrm{d}\psi.
\end{aligned}
\tag{4}
$$

The $Y^{(l)}_m$'s form an orthonormal basis set for functions defined on the unit sphere $S^2$, i.e., any square-integrable function $F(\theta,\psi) : S^2 \to \mathbb{R}$ can be expanded as $\sum_{l=0}^{\infty} \sum_{m=-l}^{l} f^{(l)}_m Y^{(l)}_m(\theta,\psi)$, and the coefficients $f^{(l)}_m$ can be calculated by $\int_0^{2\pi} \int_0^{\pi} f(\theta,\psi) Y^{(l)}_m(\theta,\psi) \sin\theta \mathrm{d}\theta \mathrm{d}\psi$. If we set $x^{(l_1)}_{m_1}, x^{(l_2)}_{m_2}$ to 0 for $l_1 > L_1, l_2 > L_2$, we actually obtain two spherical functions, $F_1(\theta,\psi) = \sum_{l_1=0}^{\infty} \sum_{m_1=-l_1}^{l_1} x^{(l_1)}_{m_1} Y^{(l_1)}_{m_1}(\theta,\psi)$ and $F_2(\theta,\psi) = \sum_{l_2=0}^{\infty} \sum_{m_2=-l_2}^{l_2} x^{(l_2)}_{m_2} Y^{(l_2)}_{m_2}(\theta,\psi)$. Thus, $(\tilde{\mathbf{x}}^{(L_1)} \otimes_{\tilde{Gaunt}} \tilde{\mathbf{x}}^{(L_2)})^{(l)}_m$ calculates the coefficients of function $F_3(\theta,\psi) = F_1(\theta,\psi) \cdot F_2(\theta,\psi)$. This perspective connects tensor products of irreps to multiplication between spherical functions, which creates new ways to accelerate the tensor product operation, as we will discuss further.

## 3.2 Fast Tensor Product using Convolution Theorem & Fast Fourier Transform

For multiplication between functions defined on the unit sphere $S^2$, a natural question is the existence of efficient computational methods. We can leverage the change of basis to achieve this goal: instead of using spherical harmonics, the same functions defined on the unit sphere $S^2$ can be equivalently represented using a 2D Fourier basis, which is known to have favorable numerical properties.

Given $F(\theta,\phi) : S^2 \to \mathbb{R}$, it can be represented by a linear combination of 2D Fourier bases as $\sum_u \sum_v f^*_{u,v} e^{i(u\theta+v\psi)}, u, v \in \mathbb{Z}$ with complex coefficients $f^*_{u,v}$ to make it a real function. To keep the convention of spherical harmonics, we also describe the 2D Fourier basis in terms of (up to) degrees $L$ if $-L \le u \le L, -L \le v \le L$. The multiplication between $F_1(\theta,\psi)$ and $F_2(\theta,\psi)$ can thus be expressed as:

$$
\begin{aligned}
F_1(\theta,\psi) \cdot F_2(\theta,\psi) &= \left( \sum_{u_1} \sum_{v_1} f^{1*}_{u_1,v_1} e^{i(u_1\theta+v_1\psi)} \right) \cdot \left( \sum_{u_2} \sum_{v_2} f^{2*}_{u_2,v_2} e^{i(u_2\theta+v_2\psi)} \right) \\
&= \sum_{u_1} \sum_{v_1} \sum_{u_2} \sum_{v_2} f^{1*}_{u_1,v_1} f^{2*}_{u_2,v_2} e^{i((u_1+u_2)\theta+(v_1+v_2)\psi)}.
\end{aligned}
\tag{5}
$$

By comparing the 2D Fourier bases, the coefficients of $F_3(\theta,\psi) = F_1(\theta,\psi) \cdot F_2(\theta,\psi)$ can be obtained by $f^{3*}_{u_3,v_3} = \sum_{u_1+u_2=u_3} \sum_{v_1+v_2=v_3} f^{1*}_{u_1,v_1} f^{2*}_{u_2,v_2}$, which is exactly the form of a 2D convolution. It is well-known that these 2D convolutions can be significantly accelerated by using the convolution theorem and Fast Fourier Transforms (FFT) (Proakis, 2007). We can then use this to develop an efficient approach to compute the tensor product of irreps in Eqn. (4).

**From spherical harmonics to 2D Fourier bases.** Since both spherical harmonics and 2D Fourier bases form an orthonormal basis set for spherical functions, we have the following equivalence: for spherical harmonics $Y^{(l)}_m(\theta,\psi), 0 \le l \le L, -l \le m \le l$, there exist coefficients $y^{l,m,*}_{u,v}$ such that $Y^{(l)}_m(\theta,\psi) = \sum_{u=-L}^{L} \sum_{v=-L}^{L} y^{l,m,*}_{u,v} e^{i(u\theta+v\psi)}$. These coefficients $y^{l,m,*}_{u,v}$ are sparse due to the orthogonality of such basis sets. That is, $y^{l,m,*}_{u,v}$ are non-zero only when $m = \pm v$.

To compute $\tilde{\mathbf{x}}^{(L_1)} \otimes_{\tilde{Gaunt}} \tilde{\mathbf{x}}^{(L_2)}$ in Eqn. (4), the first step is to convert the coefficients of spherical harmonics to coefficients of the 2D Fourier basis. From the above equivalence, we have the following conversion rule from spherical harmonics to 2D Fourier bases:

$$x_{u,v}^* = \sum_{l=0}^{L} \sum_{m=-l}^{l} x_m^{(l)} y_{u,v}^{l,m,*}, -L \leq u \leq L, -L \leq v \leq L. \tag{6}$$

**Performing 2D convolution via Fast Fourier Transform.** After converting coefficients from spherical harmonics to 2D Fourier bases, we can use the convolution theorem and Fast Fourier Transform (FFT) to efficiently compute the 2D convolution from Eqn. (5). Both $x_{u,v}^{1*}$ and $x_{u,v}^{2*}$ are transformed from the spatial domain to the frequency domain via FFT. In the frequency domain, the 2D convolution can be simply computed by element-wise multiplication. Then, the results of multiplication in the frequency domain are transformed back to the spatial domain via FFT again.

**From 2D Fourier bases to spherical harmonics.** Similarly, for 2D Fourier bases $e^{i(u\theta+v\psi)}$, where $-L \leq u \leq L$ and $-L \leq v \leq L$, there exist coefficients $z_{u,v}^{l,m,*}$ such that $e^{i(u\theta+v\psi)} = \sum_{l=0}^{L} \sum_{m=-L}^{L} z_{u,v}^{l,m,*} Y_m^{(l)}(\theta,\psi)$, and $z_{u,v}^{l,m,*}$ are also sparse, i.e., $z_{u,v}^{l,m,*}$ are non-zero only when $m = \pm v$. The last step is to convert the resulting coefficients of multiplication between spherical functions back to coefficients of spherical harmonics, which can be calculated as:

$$x_m^{(l)} = \sum_{u=-L}^{L} \sum_{v=-L}^{L} x_{u,v}^* z_{u,v}^{l,m,*}, 0 \leq l \leq L, -l \leq m \leq l. \tag{7}$$

**Computational complexity analysis.** We provide an analysis on the computational complexity of the original implementation of full tensor products of irreps of degree up to $L$. As can be seen from Eqn. (4), to calculate the $m$-th element of the output type-$l$ irrep, there exist $\mathcal{O}(L^2)$ $(l_1, l_2) \to l$ combinations of input irreps. Each combination requires a summation of $\mathcal{O}(L^2)$ multiplications. This means it requires $\mathcal{O}(L^4)$ operations for the $m$-th element of the output type-$l$ irrep. Then, retaining all $m$-elements of all output type-$l$ irreps with $0 \leq l \leq L$ has an $\mathcal{O}(L^6)$ complexity.

In our approach, we first convert coefficients of spherical harmonics to coefficients of 2D Fourier bases (see Eqn. (6)). For each $x_{u,v}^*$, since $y_{u,v}^{l,m,*}$ are non-zero only when $m = \pm v$, it requires $\mathcal{O}(L)$ operations. Thus, the first step of our approach requires $\mathcal{O}(L^3)$ operations in total. For the second step, the 2D convolution in Eqn. (5) has an $\mathcal{O}(L^2 \log L)$ complexity using FFT. Similarly, the last step of our approach also only requires $\mathcal{O}(L^3)$, due to the sparsity of $z_{u,v}^{l,m,*}$. Therefore, our approach has an $\mathcal{O}(L^3)$ complexity to compute the full tensor products of irreps of degrees up to $L$, which achieves substantial acceleration compared to the original implementation.

## 3.3 Designing Efficient Equivariant Operations via the Gaunt Tensor Product

Based on the approach in Section 3.2, we introduce the Gaunt Tensor Product, a technique that efficiently computes tensor products of irreps utilizing Gaunt coefficients. Our Gaunt Tensor Product can be applied across different model architectures in the literature, and serves as a new method for efficient equivariant operations. We provide a comprehensive study on major equivariant operation classes (*Equivariant Feature Interactions*, *Equivariant Convolutions*, and *Equivariant Many-body Interactions*) and show how to design efficient counterparts via our Gaunt Tensor Product. We present additional details in Appendix C.

**Equivariant Feature Interactions.** Given two features $\tilde{\mathbf{x}}^{(L_1)}, \tilde{\mathbf{y}}^{(L_2)}$ containing irreps of up to degree $L_1$ and $L_2$ respectively, the operations in this class are commonly in the following form: $(\tilde{\mathbf{x}}^{(L_1)} \otimes_{cg}^{\mathbf{w}} \tilde{\mathbf{y}}^{(L_2)})^{(l)} = \sum_{l_1=0}^{L_1} \sum_{l_2=0}^{L_2} w_{l_1,l_2}^l (\mathbf{x}^{(l_1)} \otimes_{cg} \mathbf{y}^{(l_2)})^{(l)}$, where $w_{l_1,l_2}^l \in \mathbb{R}$ are the learnable weights for each $(l_1, l_2) \to l$ combination. Compared to Eqn. (4), our Gaunt Tensor Product is exactly in the same form, except for the learnable weights $w_{l_1,l_2}^l$. Our approach can be naturally extended by reparameterizing $w_{l_1,l_2}^l$ as $w_{l_1} \cdot w_{l_2} \cdot w_l$, where $w_{l_1}, w_{l_2}, w_l \in \mathbb{R}$. The weighted combinations can be equivalently achieved in Eqn. (4) by separately multiplying $w_{l_1}, w_{l_2}, w_l$ with $\mathbf{x}^{(l_1)}, \mathbf{x}^{(l_2)}$ and $(\mathbf{x}^{(l_1)} \otimes_{Gaunt} \mathbf{x}^{(l_2)})^{(l)}$ correspondingly, in which case the tensor product of irreps can be still efficiently computed by using the approach in Section 3.2.

These operations facilitate interactions between irreps from different dimensional spaces and objects, which have been widely used in recent work. For example, So3krates (Frank et al., 2022) developed

the atomic interaction layer, where both $\tilde{\mathbf{x}}^{(L_1)}$ and $\tilde{\mathbf{y}}^{(L_2)}$ are the equivariant features $\mathcal{X}_i^{(L)}$ of atom $i$. Additionally, in the Quantum Hamiltonian prediction task, such operations are commonly used to interact features for constructing Hamiltonian matrices (Unke et al., 2021; Yu et al., 2023b). For instance, QHNet (Yu et al., 2023b) instantiates $\tilde{\mathbf{x}}^{(L_1)}$ and $\tilde{\mathbf{y}}^{(L_2)}$ as equivariant features of atom pairs $(\hat{\mathbf{x}}_i, \hat{\mathbf{x}}_j)$ and uses the output features as the input of the non-diagonal matrix elements prediction block.

**Equivariant Convolutions.** One special case of Equivariant Feature Interaction is to instantiate $\tilde{\mathbf{y}}^{(L_2)}$ as spherical harmonics filters, i.e., $\sum_{l_1=0}^{L_1} \sum_{l_2=0}^{L_2} h_{l_1,l_2}^l (\mathbf{x}_i^{(l_1)} \otimes_{cg} Y^{(l_2)}(\frac{\mathbf{r}_i - \mathbf{r}_j}{||\mathbf{r}_i - \mathbf{r}_j||}))^{(l)}$, where $\mathbf{r}_i, \mathbf{r}_j \in \mathbb{R}^3$ are positions of objects, $Y^{(l_2)} : S^2 \to \mathbb{R}^{2l_2+1}$ is the type-$l_2$ spherical harmonic, and $h_{l_1,l_2}^l \in \mathbb{R}$ are learnable weights calculated based on the relative distance and types of objects $i, j$. Similarly, the $h_{l_1,l_2}^l$ can be reparameterized for our approach. Passaro & Zitnick (2023) provide an interesting observation that if we select a rotation $g \in SO(3)$ with $\mathbf{D}^{(1)}(g)\frac{\mathbf{r}_i - \mathbf{r}_j}{||\mathbf{r}_i - \mathbf{r}_j||} = (0, 1, 0)$, then $Y_m^{(l)}(\mathbf{D}^{(1)}(g)\frac{\mathbf{r}_i - \mathbf{r}_j}{||\mathbf{r}_i - \mathbf{r}_j||}) \propto \delta_m^{(l)}$, i.e., is non-zero only if $m = 0$. This sparsification further propagates to the coefficients of the 2D Fourier basis, which brings additional acceleration in Eqn. (6).

Equivariant Convolutions are the basic operations for equivariant message passing (Thomas et al., 2018; Fuchs et al., 2020; Brandstetter et al., 2022; Batzner et al., 2022; Liao & Smidt, 2023; Passaro & Zitnick, 2023). The spherical harmonics filters encode the geometric relations between objects and interact with the object features in an equivariant manner, which plays an essential role in learning semantics and geometric structures under equivariance constraints.

**Equivariant Many-body Interactions.** Operations in this class perform multiple tensor products among equivariant features, i.e., $\tilde{\mathbf{x}}_1^{(L_1)} \otimes_{\tilde{c}g} \tilde{\mathbf{x}}_2^{(L_2)} \otimes_{\tilde{c}g} ... \otimes_{\tilde{c}g} \tilde{\mathbf{x}}_n^{(L_n)}$. Such operations are commonly used for constructing many-body interaction effects (Drautz, 2019; Batatia et al., 2022a;b; Nigam et al., 2022; Kovács et al., 2023), which can improve performance for ML force fields. For example, given equivariant features $A_i^{(L')}$ of atom $i$, MACE (Batatia et al., 2022b) constructs the many-body features by $\sum_{\eta_\nu} \sum_{\mathbf{lm}} C_{\eta_\nu,\mathbf{lm}}^{LM} \prod_{\xi=1}^{\nu} A_{i,m_\xi}^{(l_\xi)}$, which is the explicit form of performing tensor products with $A_i^{(L)}$ itself $\nu$ times, i.e., $B_{\nu,i} = A_i^{(L')} \otimes_{\tilde{c}g} ... \otimes_{\tilde{c}g} A_i^{(L')}$. Similarly, such operations can be equivalently viewed as multiplications of multiple spherical functions. Using this, we can use a divide-and-conquer method to efficiently compute multiple 2D convolutions for further speedups.

**Discussion.** It is common in models to maintain $C$ channels of equivariant features, i.e., $\tilde{\mathbf{x}}_{i,c}^{(L)}, 0 \le c < C$ contains irreps of degrees up to $L$. This extension can be naturally included in our approach by defining the combination rules of features in different channels, e.g., channel-wise ($\mathcal{O}(C)$ ops) or channel-mixing ($\mathcal{O}(C^2)$ ops). Note that our approach is not limited to full tensor products only. In fact, given $\mathbf{x}^{(l_1)} \otimes_{cg} \mathbf{y}^{(l_2)}$ only, the approach presented in Section 3.2 can still be used, which reduces $\mathcal{O}(\tilde{L}^3)$ to $\mathcal{O}(\tilde{L}^2 \log \tilde{L})$ with $\tilde{L} = l_1 + l_2$. Overall, we demonstrate the generality of our Gaunt Tensor Product, and we believe that it is a starting point with more possibilities to explore in the future.

## 4 EXPERIMENTS

We empirically study the efficiency and effectiveness of our Gaunt Tensor Product. First, we provide comprehensive efficiency comparisons between our Gaunt Tensor Product and the original implementations of the operation classes mentioned in Section 3.3. Second, we perform a sanity check to investigate the effects brought by the different parameterizations of our Gaunt Tensor Product versus the Clebsch-Gordan Tensor Product. Finally, we present results applying our Gaunt Tensor Product to various model architectures and task types. Experiment details are presented in Appendix E. The code will be released at https://github.com/lsj2408/Gaunt-Tensor-Product.

**Efficiency Comparisons.** (1) For the Equivariant Feature Interaction operation, we use the e3nn implementation (Geiger & Smidt, 2022) as the baseline. We randomly sample 10 pairs of features containing irreps of up to degree $L$ with 128 channels. The average inference time is reported. (2) For the Equivariant Convolution operation, we use the eSCN implementation (Passaro & Zitnick, 2023) as the baseline, due to its enhanced efficiency over e3nn for this operation. In this setting, pairs of features and spherical harmonics filters are sampled instead. We implement our Gaunt Tensor Product by using the insights of eSCN for further acceleration (Section 3.3). (3) For the Equivariant Many-body Interaction operation, we compare our approach with both the e3nn and the MACE implementations (Batatia et al., 2022b). We evaluate various degrees of $L$ and the number of tensor product operands $\nu$.

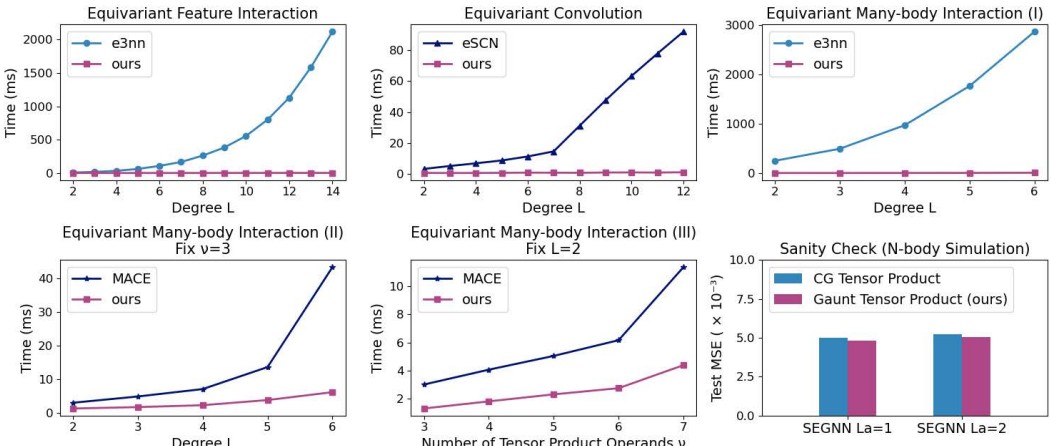

Figure 1: Results on efficiency comparisons and sanity check. We comprehensively compare our Gaunt Tensor Product with implementations of e3nn, eSCN, and MACE on corresponding equivariant operation classes. In all settings, our approach achieves significant speedups. Our Gaunt Tensor Product parameterization further passes the sanity check with SEGNN on the N-body simulation task.

|  |  |  |  | | S2EF | | |
|---|---|---|---|---|---|---|---|
| | | | | Energy MAE | Force MAE | Force Cos | EFwT |
| **Model** | | | | [meV] ↓ | [meV/Å] ↓ | ↑ | [%] ↑ |
| SchNet (Schütt et al., 2018) | | | | 1400 | 78.3 | 0.109 | 0.00 |
| DimeNet++ (Gasteiger et al., 2020a) | | | | 805 | 65.7 | 0.217 | 0.01 |
| SpinConv (Shuaibi et al., 2021) | | | | 406 | 36.2 | 0.479 | 0.13 |
| GemNet-dT (Gasteiger et al., 2021) | | | | 358 | 29.5 | 0.557 | 0.61 |
| GemNet-OC (Gasteiger et al., 2022) | | | | 286 | 25.7 | 0.598 | 1.06 |
| | $L$ | # layers | # batch | | | | |
| SCN 1-tap 1-band (Zitnick et al., 2022) | 6 | 12 | 64 | 299 | 24.3 | 0.605 | 0.98 |
| SCN 4-tap 2-band (Zitnick et al., 2022) | 6 | 12 | 64 | 279 | 22.2 | 0.643 | 1.41 |
| eSCN (Passaro & Zitnick, 2023) | 4 | 12 | 96 | 291 | 22.2 | 0.637 | 1.39 |
| eSCN (Passaro & Zitnick, 2023) | 6 | 12 | 96 | 294 | 21.3 | 0.653 | 1.45 |
| EquiformerV2 (Liao et al., 2024) | 4 | 12 | 64 | 284 | 21.4 | 0.657 | 1.51 |
| EquiformerV2 (Liao et al., 2024) | 6 | 12 | 64 | 285 | 20.5 | 0.663 | 1.67 |
| EquiformerV2 + Gaunt-Selfmix (ours) | 4 | 12 | 64 | 277 | 20.9 | 0.661 | 1.82 |
| EquiformerV2 + Gaunt-Selfmix (ours) | 6 | 12 | 64 | **276** | **20.1** | **0.669** | **1.95** |

OC20 2M Validation

Table 1: Results on the OC20 S2EF task. The results are averaged across the four OC20 Validation set splits. Bolded values denote the best performance.

The results are presented in Fig.1. Our Gaunt Tensor Product achieves substantial acceleration compared to the e3nn implementation, e.g., multiple orders of magnitude speed-up when $L>7$. Compared to the efficient eSCN implementation, our approach has further acceleration, due to the combination of these two methods. Finally, our Gaunt Tensor Product consistently outperforms the MACE implementation across different $L$ and $\nu$. These results demonstrate the improved efficiency of our approach.

**Sanity Check.** As shown in Eqn. (3), the Gaunt coefficients are proportional to the Clebsch-Gordan coefficients for the constants, which leads to different parameterizations of our Gaunt Tensor Product and the Clebsch-Gordan Tensor Product. To investigate the underlying effects, we conduct a sanity check by applying the parameterization of the Gaunt Tensor Product to the SEGNN (Brandstetter et al., 2022) model on the N-body simulation task (Satorras et al., 2021), which requires the model to forecast the positions of a set of particles modeled by predefined interaction rules. Following Brandstetter et al. (2022), we employ a 4-layer SEGNN and keep all hyperparameters the same to compare the performance of different parameterizations.

In the last panel of Figure 1, we see that SEGNN with our Gaunt Tensor Product performs competitively compared to the Clebsch-Gordan Tensor Product implementation, which demonstrates that our parameterization does not hurt the performance of equivariant operations.

Table 2: Results on the 3BPA dataset. Energy (E, meV) and force (F, meV/Å) errors are evaluated. Standard deviations over three runs are reported if available. Bold values denote the best efficiency.

| | | ACE | sGDML | Allegro | NequIP | BOTNet | MACE | MACE-Gaunt |
|---|---|---|---|---|---|---|---|---|
| 300 K | E | 7.1 | 9.1 | 3.84 (0.08) | 3.3 (0.1) | 3.1 (0.13) | 3.0 (0.2) | 2.9 (0.1) |
| | F | 27.1 | 46.2 | 12.98 (0.17) | 10.8 (0.2) | 11.0 (0.14) | 8.8 (0.3) | 9.2 (0.1) |
| 600 K | E | 24.0 | 484.8 | 12.07 (0.45) | 11.2 (0.1) | 11.5 (0.6) | 9.7 (0.5) | 10.6 (0.5) |
| | F | 64.3 | 439.2 | 29.17 (0.22) | 26.4 (0.1) | 26.7 (0.29) | 21.8 (0.6) | 22.2 (0.2) |
| 1200 K | E | 85.3 | 774.5 | 42.57 (1.46) | 38.5 (1.6) | 39.1 (1.1) | 29.8 (1.0) | 30.4 (1.2) |
| | F | 187.0 | 711.1 | 82.96 (1.77) | 76.2 (1.1) | 81.1 (1.5) | 62.0 (0.7) | 63.1 (1.2) |
| Dihedral Slices | E | - | - | - | - | 16.3 (1.5) | 7.8 (0.6) | 9.9 (0.3) |
| | F | - | - | - | - | 20.0 (1.2) | 16.5 (1.7) | 17.7 (1.1) |
| Speed-ups (v.s. e3nn) | | - | - | - | - | - | 33.2x | **43.7x** |
| Memory costs (v.s. e3nn) | | - | - | - | - | - | 32.8% | **5.8%** |

**OC20 S2EF performance.** Following Gasteiger et al. (2021); Zitnick et al. (2022); Passaro & Zitnick (2023), we evaluate our approach on the large-scale Open Catalyst 2020 (OC20) dataset, which consists of 1.2M DFT relaxations. Each data instance in OC20 has an adsorbate molecule placed on a catalyst surface. The core task is Structure-to-Energy-Forces (S2EF), which requires the model to predict the energy and per-atom forces of the adsorbate-catalyst complex. Recently, the EquiformerV2 model (Liao et al., 2024) achieved state-of-the-art performance on the OC20 S2EF task, which uses the efficient eSCN implementation to build a scalable Transformer backbone. However, this implementation restricts the available tensor product operations to Equivariant Convolutions only.

We apply our Gaunt Tensor Product to construct an efficient Equivariant Feature Interaction operation, Selfmix, and add it to each layer of the EquiformerV2 model. Our approach retains efficiency even with such modifications. Following Liao et al. (2024), we train 12-layer EquiformerV2 models and keep all configurations the same. We use the OC20 2M subset for training. The results are presented in Table 1. We show that EquiformerV2 with our Gaunt Selfmix operation achieves better performance on the S2EF task, i.e., 16.8% relative improvement on the EFwT metric with $L$=6.

**3BPA performance.** Following Batzner et al. (2022); Musaelian et al. (2023); Batatia et al. (2022b), we use the 3BPA dataset (Kovács et al., 2021) to benchmark the Equivariant Many-body Interaction operations. The training set of 3BPA contains 500 geometries sampled from the 300 K molecular dynamics simulation of the large and flexible drug-like molecule 3-(benzyloxy)pyridin-2-amine (3BPA). Both in- (300 K) and out-of-distribution test sets (600 K, 1200 K, dihedral slices) are used to evaluate the model's ability to predict the energy and forces of the geometrical structure. We apply our Gaunt Tensor Product to the MACE architecture, which uses equivariant features containing irreps of up to degree $L = 2$. The Equivariant Many-body Interaction operation in MACE performs two tensor products among three equivariant features. We keep all training configurations the same as in Batatia et al. (2022b). The details of baselines and settings are presented in Appendix E.4.

The results are presented in Table 2. MACE, integrated with our Gaunt Tensor Product, performs competitively compared to its original implementation, which has the best performance among competitive baselines. This affirms that our parameterization in the Equivariant Many-body Interaction does not hurt accuracy. MACE with our Gaunt Tensor Product achieves significant acceleration compared to both the e3nn implementation and the original MACE implementation, i.e., 43.7x speed-ups compared to the baselines. Note that the MACE implementation trades space for speed and incurs additional memory costs, while our approach has low memory costs that are friendly for deployment, e.g., reducing 82.3% relative memory costs versus MACE.

## 5 CONCLUSION

We introduce a systematic approach, the Gaunt Tensor Product, to significantly accelerate the computation of the tensor products of irreps. Our Gaunt Tensor Product is established through the connection between Clebsch-Gordan coefficients and Gaunt coefficients, which enables the equivalent expression of the tensor product operation as the multiplication between spherical functions represented by spherical harmonics. This viewpoint allows us to change the basis for the equivariant operations to a 2D Fourier basis. Multiplication between functions in this basis can thus be accelerated via the convolution theorem and Fast Fourier Transforms. Our experiments demonstrate both the improved efficiency and effectiveness of our approach. Overall, our Gaunt Tensor Product provides new opportunities for efficient equivariant operations, opening avenues for exploration in the future.

**Acknowledgements.** This work acknowledges support from Laboratory Directed Research and Development (LDRD) funding under Contract Number DE-AC02-05CH11231. We thank Nithin Chalapathi, Di He, Eric Qu, Danny Reidenbach, and Bohang Zhang for helpful discussions and feedback. We also thank all reviewers for their valuable suggestions.

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

## A BACKGROUND

### A.1 GROUP THEORY

**Groups.** Let $G$ denote a group, which is a non-empty set together with a binary operation denoted as $\circ : G \times G \to G$. The group operation $\circ$ satisfies the following properties known as *group axioms*:

- *Associativity*: For all $a, b, c \in G$, we have $a \circ (b \circ c) = (a \circ b) \circ c = a \circ b \circ c$.
- *Identity*: For all $a \in G$, there exists an element $e \in G$ such that $a \circ e = e \circ a = a$.
- *Inverse*: For each $a \in G$, there exists an element $a^{-1} \in G$ such that $a^{-1} \circ a = a \circ a^{-1} = e$.
- *Closure*: For all $a, b \in G$, $a \circ b \in G$.

For brevity, the group operation $\circ$ can be omitted and $a \circ b$ can be abbreviated as $ab$. Groups can be finite or infinite, countable or uncountable, compact or non-compact. The Groups are algebraic structures that are widely used across physics (Cornwell, 1997), chemistry (Cotton, 1991), materials science (Burns, 2014), and other scientific areas.

**Group Representations.** As introduced above, the group operation $\circ$ describes how the group elements interact with each other. To describe how the group elements act on vector spaces, *group representations* are used. Formally, a representation of a group $G$ on a vector space $V$ is a mapping $\rho^V : G \to GL(V)$. Here, $GL(V)$ denotes the general linear group on vector space $V$, which is the set of invertible matrices together with the binary operation of ordinary matrix multiplication. $V$ is called the representation space and the dimension of $V$ is called the dimension of the representation. The mapping $\rho^V(g)$ parameterized by the group element $g \in G$ is thus an invertible linear transformation on $V$, which follows the group structure (also known as *group homomorphism*[1]) that $\rho(g_1 g_2)v = \rho(g_1)\rho(g_2)v$ for all $g_1, g_2 \in G, v \in V$.

Assuming $V$ is the finite-dimensional real vector space $\mathbb{R}^d$, the group representations are invertible matrices in $\mathbb{R}^{d \times d}$. Let $\mathbf{D}(g)$ denote the representation of $g$ in the matrix form. Representations of the rotation group in the three-dimensional Euclidean space are matrices $\mathbf{R} \in \mathbb{R}^{3 \times 3}, \det(\mathbf{R}) = 1$, and the representations act on $\mathbf{x} \in \mathbb{R}^d$ via matrix multiplications $\mathbf{D}(g)\mathbf{x} = \mathbf{R}\mathbf{x}$.

For any two representations, we say $\mathbf{D}(g)$ and $\mathbf{D}'(g)$ are equivalent if there exists a similarity transform such that:

$$\mathbf{D}'(g) = \mathbf{Q}^{-1}\mathbf{D}(g)\mathbf{Q}. \tag{8}$$

That is, both representations $\mathbf{D}'(g)$ and $\mathbf{D}(g)$ represent the same group element $g$ with different basis, and the change of basis is carried out by $\mathbf{Q}$.

**Examples of Groups.** While the Group is an abstract mathematical concept, there are numerous classes of objects and transformations that have group structures. In this paper, we are interested in the group of translations, rotations, and reflections in the three-dimensional Euclidean space, which is known as the 3D Euclidean group and denoted as $E(3)$. For each element $g$ in the $E(3)$ group, its representation on $\mathbb{R}^3$ can be parameterized by pairs of translation vectors $\mathbf{t} \in \mathbb{R}^3$ and orthogonal transformation matrices $\mathbf{R} \in \mathbb{R}^{3 \times 3}, \det(\mathbf{R}) = \pm 1$, i.e., $\rho^{\mathbb{R}^3}(g) = (t, \mathbf{R})$. Given a vector $\mathbf{x} \in \mathbb{R}^3$, we have $\rho^{\mathbb{R}^3}(g)[x] := \mathbf{R}\mathbf{x} + \mathbf{t}$. The group operation $\circ$ and inverse of $E(3)$ are defined by $g \circ g' := (\mathbf{R}\mathbf{t}' + \mathbf{t}, \mathbf{R}\mathbf{R}')$ and $g^{-1} := (\mathbf{R}^{-1}\mathbf{x}, \mathbf{R}^{-1})$, where $g = (\mathbf{t}, \mathbf{R}), g' = (\mathbf{t}', \mathbf{R}') \in E(3)$.

The $E(3)$ group is a semi-direct product of the group of translations $\mathbb{R}^3$ with the group of orthogonal transformations $O(3)$, i.e., $E(3) = \mathbb{R}^3 \rtimes O(3)$. Thus, the group elements in $E(3)$ can be decomposed in an $O(3)$-transformation (rotation and/or reflection) followed by a translation. Note that translations are easy to deal with in modern neural networks. Therefore, we focus mainly on the orthogonal group $O(3)$, which consists of rotations and reflections. For each element $g$ in the $O(3)$ group, its representation on $\mathbb{R}^3$ is thus parameterized by orthogonal transformation matrices $\mathbf{R} \in \mathbb{R}^{3 \times 3}, \det(\mathbf{R}) = \pm 1$, i.e., $g = \mathbf{R}$. Similarly, the group operation $\circ$ and inverse of $O(3)$ are defined by $g \circ g' := \mathbf{R}\mathbf{R}'$ and $g^{-1} := \mathbf{R}^{-1}$, where $g = \mathbf{R}, g' = \mathbf{R}' \in O(3)$. The special orthogonal group $SO(3)$ has the same group operation and inverse, but excludes reflections. Thus, each element $g = \mathbf{R} \in SO(3)$ has $\det(\mathbf{R}) = 1$.

---

[1]Given two groups $(G, \circ_G)$ and $(H, \circ_H)$, a function mapping $\pi : G \to H$ is said to be a *group homomorphism* if for all $g, h \in G$ we have $\pi(g) \circ_H \pi(h) = \pi(g \circ_G h)$

## A.2 EQUIVARIANCE

Formally, let $\phi : \mathcal{X} \to \mathcal{Y}$ denote a function mapping between vector spaces. Given a group $G$, let $\rho^{\mathcal{X}}$ and $\rho^{\mathcal{Y}}$ denote its group representations on $\mathcal{X}$ and $\mathcal{Y}$ respectively. A function $\phi : \mathcal{X} \to \mathcal{Y}$ is said to be equivariant if it satisfies the following condition:

$$\rho^{\mathcal{Y}}(g)[\phi(x)] = \phi(\rho^{\mathcal{X}}(g)[x]), \text{ for all } g \in G, x \in \mathcal{X}. \tag{9}$$

When $\rho^{\mathcal{Y}} = \mathcal{I}^{\mathcal{Y}}$ for all $g \in G$, it is a special case of equivariance which is known as invariance, i.e., a function $\phi : \mathcal{X} \to \mathcal{Y}$ is said to be invariant if it satisfies the following condition:

$$\phi(x) = \phi(\rho^{\mathcal{X}}(g)[x]), \text{ for all } g \in G, x \in \mathcal{X}. \tag{10}$$

Intuitively, an equivariant function mapping transforms the output predictably in response to transformations on the input, whereas an invariant function mapping produces an output that remains unchanged by transformations applied to the input. For example, the total energy of a molecular system is invariant to the rotations, translations, and reflections acted on each position of atoms, i.e., the energy function with respect to the atom positions is an invariant function to the $E(3)$ group. If a rotation is applied to each position of atoms, the direction of the force on each atom will also rotate, i.e., the force function is an equivariant function to the $SO(3)$ group.

## A.3 IRREDUCIBLE REPRESENTATIONS

**Subrepresentations and Irreducible Representations.** Let $W$ be a subspace of $V$. When $W$ is invariant under the group $G$, i.e., $\rho^W(g)x \in W$ for all $x \in W$ and $g \in G$, the representation $\rho^W$ of the group $G$ on the vector space $W$ is called a *subrepresentation* of $\rho^V$. The subspace $W$ is also called *G-invariant*. For any representation $\rho^V$ of a group G, it has two *trivial subrepresentations*: (1) $\rho^V$ itself; (2) the zero representation, both of which satisfy the above definition.

After defining the concept of subrepresentations, we are ready to introduce *irreducible representations*, which plays an essential role in modern rotationally equivariant neural networks. A representation $\rho^V : G \to GL(V)$ is said to be *irreducible* if it has only trivial subrepresentations. If there is a proper nontrivial $G$-invariant subspace, $\rho^V$ is said to be *reducible*.

**Wigner-D matrices.** We introduce the Wigner-D matrices, which serve as the irreducible representations of the rotation group $SO(3)$. For vector spaces of dimension $2l + 1, l = 0, 1, 2, ...$, there exists a collection of representations indexed with $l$ for each element in the $SO(3)$ group, which are called *Wigner-D matrices* and denoted as $\mathbf{D}^{(l)}(g)$ for $g \in SO(3)$. Each Wigner-D matrix $\mathbf{D}^{(l)}(g)$ is the irreducible representation of the $SO(3)$ group on subspace with dimension $2l + 1$, and any matrix representation $\mathbf{D}(g)$ of $SO(3)$ on vector space $V$ can be *reduced* to an equivalent block diagonal matrix representation with Wigner-D matrices along the diagonal:

$$\mathbf{D}(g) = \mathbf{Q}^{-1}(\mathbf{D}^{(l_1)}(g) \oplus \mathbf{D}^{(l_2)}(g) \oplus ...)\mathbf{Q} = \mathbf{Q}^{-1} \begin{pmatrix} \mathbf{D}^{(l_1)}(g) & & \\ & \mathbf{D}^{(l_2)}(g) & \\ & & ... \end{pmatrix} \mathbf{Q}. \tag{11}$$

Here, $\mathbf{Q}$ is the change of basis that makes $\mathbf{D}(g)$ and the block diagonal matrix representation equivalent. This equivalence provides an alternative perspective that the representations that act on the vector space $V$ are made up of each Wigner-D matrix $\mathbf{D}^{(l_i)}$, which only acts on a subspace $V_{l_i}$ with dimension $2l_i + 1$ of $V$. Then, the vector space $V$ can be factorized into $V_{l_1} \oplus V_{l_2} \oplus .....$ By convention, the space $V_{l_i}$ is called a *type-$l_i$ vector space* and $\mathbf{x} \in V_{l_i}$ is called a *type-$l_i$ vector*, which is transformed by the Wigner-D matrix $\mathbf{D}^{(l_i)}$. In Appendix A.1, we have already introduced the Wigner-D matrix in the type-1 space, i.e., $\mathbf{D}^{(1)} = \mathbf{R} \in \mathbb{R}^{3 \times 3}, \det(\mathbf{R}) = 1$. The type-0 space is $\mathbb{R}$ and is invariant to the $SO(3)$ group, i.e., $\mathbf{D}^{(0)}(g)x = x$ for all $g \in G$ and $x \in \mathbb{R}$.

As previously mentioned, we are interested in the $E(3)$ group consisting of translations, rotations, and reflections. The Wigner-D matrix representations can be naturally extended to the $O(3)$ group by further including the reflections via a direct product, while the translations can be easily dealt with in modern equivariant networks. We refer the readers to Commins (1995); Wigner (2012) for the explicit forms of the Wigner-D matrices and the e3nn[2] codebase (Geiger & Smidt, 2022) for the numerical implementation.

---

[2]https://github.com/e3nn/e3nn

## A.4 SPHERICAL HARMONICS

Spherical harmonics are a class of functions defined on the unit sphere $S^2$. These functions form an orthonormal basis set, and thus every function defined on $S^2$ can be expressed as a series of spherical harmonics, which is similar to the Fourier series. By convention, we use $Y_m^{(l)} : S^2 \to \mathbb{R}, -l \le m \le l, l = 0, 1, ...$ to denote the real spherical harmonics, where $l$ and $m$ denote the degrees and orders of spherical harmonics respectively. Both spherical coordinates $(\theta, \psi), \theta \in [0, \pi], \psi \in [0, 2\pi)$ and Cartesian coordinates $\mathbf{r} \in \mathbb{R}^3, ||\mathbf{r}|| = 1$ are widely used to express the points on the sphere $S^2$ as the input variables of the spherical harmonics. Next, we will introduce several important properties of the spherical harmonics used in equivariant networks, and their connections to the irreducible representations of the $SO(3)$ group.

**Spherical harmonics expansion.** Formally, on the unit sphere $S^2$, any square-integrable function $f : S^2 \to \mathbb{R}$ can be expanded as a linear combination of spherical harmonics:

$$f(\theta, \psi) = \sum_{l=0}^{\infty} \sum_{m=-l}^{l} f_m^{(l)} Y_m^{(l)}(\theta, \psi). \tag{12}$$

This expansion holds in the sense of mean-square convergence:

$$\lim_{L \to \infty} \int_0^{2\pi} \int_0^{\pi} |f(\theta, \psi) - \sum_{l=0}^{L} \sum_{m=-l}^{l} f_m^{(l)} Y_m^{(l)}(\theta, \psi)| \sin\theta d\theta d\psi = 0. \tag{13}$$

Similar to the Fourier series, the spherical harmonic coefficients $f_m^{(l)}$ can be obtained by using the orthonormal property of spherical harmonics:

$$f_m^{(l)} = \int_0^{2\pi} \int_0^{\pi} f(\theta, \psi) Y_m^{(l)}(\theta, \psi) d\theta \sin\theta d\theta d\psi. \tag{14}$$

**Equivariance of spherical harmonics.** The spherical harmonics have various intriguing properties, especially with respect to the equivariance of some groups consisting of transformations of interest, e.g., rotations and reflections:

- **Rotations**. Given a rotation $g \in SO(3)$ and its representation on $\mathbb{R}^3$ as $\rho^{\mathbb{R}^3}(g) = \mathbf{R} \in \mathbb{R}^{3 \times 3}, \det(\mathbf{R}) = 1$, the unit vector $\mathbf{r} \in \mathbb{R}^3, ||\mathbf{r}|| = 1$ is rotated to $\mathbf{r}' = \mathbf{R}\mathbf{r}$. Then we have:

$$Y_m^{(l)}(\mathbf{R}\mathbf{r}) = Y_m^{(l)}(\mathbf{r}') = \sum_{m'=-l}^{l} \mathbf{D}_{mm'}^{(l)}(g) Y_{m'}^{(l)}(\mathbf{r}). \tag{15}$$

Let $Y^{(l)}(\mathbf{r}) = [Y_{-l}^{(l)}(\mathbf{r}), ..., Y_l^{(l)}(\mathbf{r})]^\top$, then the above property can be abbreviated as,

$$Y^{(l)}(\mathbf{R}\mathbf{r}) = \mathbf{D}^{(l)}(g) Y^{(l)}(\mathbf{r}), \text{ for all } g \in SO(3), \mathbf{r} \in S^2, \tag{16}$$

where $\mathbf{D}^l(g)$ denotes the matrix representation of the SO(3) group element $g$ on the $2l+1$ vector space, and $\mathbf{D}_{mm'}^{(l)}(g)$ corresponds to the $(m, m')$-th matrix element.

- **Reflections**. The reflection transformation (also called inversion) is represented by the operator $P\Phi(\mathbf{r}) = \Phi(-\mathbf{r})$. Then, we have:

$$Y_m^{(l)}(-\mathbf{r}) = (-1)^l Y_m^{(l)}(\mathbf{r}). \tag{17}$$

**Connections between spherical harmonics and irreducible representations of $SO(3)$.** The equivariance of spherical harmonics indicates that there are strong connections between this function class and group representation theory. The $Y_m^{(l)}$'s of degree $l$ form a basis set of functions for the irreducible representation of the $SO(3)$ group on the $2l+1$ vector space (Kosmann-Schwarzbach et al., 2010). Each irreducible representation of $SO(3)$ can be realized in a finite-dimensional Hilbert space of functions on the sphere, the restrictions of harmonic homogeneous polynomials[3] of a given

---

[3]Given a multivariate polynomial $P$ over a field, if the polynomial $p$ satisfies: (1) $\Delta P = 0$; (2) $P(\lambda x_1, ..., \lambda x_n) = \lambda^d P(x_1, ..., x_n)$, it is called a *harmonic homogenous polynomial*.

degree, and such a representation is unitary. The orthonormal basis of this space is exactly the introduced spherical harmonics. From this perspective, the spherical harmonics are proportional to the Wigner-D matrix elements: $Y_m^{(l)}(\mathbf{r}) \propto \frac{1}{\sqrt{2l+1}} \mathbf{D}_{m0}^{(l)}(g)$ given that the rotation $g$ is parameterized by Euler angles (Pio, 1966) $(\alpha, \beta, \gamma)$, and the sphere point $\mathbf{r}$ is obtained by using the rotation $g'$ parameterized by $(\alpha, \beta, 0)$ on the $x$-unit vector. By definition, the spherical harmonics are invariant to the $\gamma$-rotation, and the relations to the Wigner-D matrix elements show the connections to the irreducible representations of the $SO(3)$ group.

## A.5 Tensor Product of Representations

In the previous subsections, we defined the group representations on vector spaces. To map the representations between vector spaces, the tensor product operation provides a principled approach, with useful properties for constructing equivariant networks. Formally, given two vectors $\mathbf{x} = [x_1, ..., x_{d_1}]^\top \in \mathbb{R}^{d_1}, \mathbf{y} = [y_2, ..., y_{d_2}]^\top \in \mathbb{R}^{d_2}$, the tensor product between $\mathbf{x}$ and $\mathbf{y}$ follows $\mathbf{x} \otimes \mathbf{y} = \mathbf{x}\mathbf{y}^\top \in \mathbb{R}^{d_1 \times d_2}$. Given two representations $\pi_1 : G \to GL(V_1)$ and $\pi_2 : G \to GL(V_2)$ on vector spaces $V_1$ and $V_2$ respectively, the tensor product of representations is given by $\pi_1 \otimes \pi_2 : G \to GL(V_1 \otimes V_2)$, i.e., $\pi_1 \otimes \pi_2(g) = \pi_1(g) \otimes \pi_2(g)$, where $\pi_1(g) \otimes \pi_2(g)$ is the tensor product of linear maps previously defined. This operation maps representations on spaces $V_1, V_2$ to the $V_1 \otimes V_2$ space with dimension being $(2l_1 + 1) \times (2l_2 + 1)$. Then, the equivariance of the tensor product operation to the $SO(3)$ group is natural to show:

$$(\mathbf{D}^{(l_1)}(g) \otimes \mathbf{D}^{(l_2)}(g))(\mathbf{x}_1 \otimes \mathbf{x}_2) = (\mathbf{D}^{(l_1)}(g)\mathbf{x}_1) \otimes (\mathbf{D}^{(l_2)}(g)\mathbf{x}_2), \text{ for all } \mathbf{x}_1 \in V_1, \mathbf{x}_2 \in V_2, \quad (18)$$

where $\mathbf{D}^{(l_1)}(g) \otimes \mathbf{D}^{(l_2)}(g)$ is a matrix representation of $g \in SO(3)$ in the vector space $V_1 \otimes V_2$.

**Clebsch-Gordan Tensor Product.** As the tensor product of representations is still a representation on the corresponding vector space, we can further decompose it into a direct sum of irreducible representations, as shown in Eqn. (11). Then, the resulting vector of the tensor product can be organized via a change of basis into parts that are individually transformed by Wigner-D matrices of different types, i.e. $\mathbf{x}_1 \otimes \mathbf{x}_2 \in V = V_0 \oplus V_1 \oplus ... \oplus V_l \oplus ...$, with $V_l$ being the type-$l$ vector space.

The Clebsch-Gordan Tensor Product provides a way to explicitly calculate the resulting type-$l$ vector after the direct sum decomposition. Given $\mathbf{x}^{(l_1)} \in \mathbb{R}^{2l_1+1}$ and $\mathbf{x}^{(l_2)} \in \mathbb{R}^{2l_2+1}$, the $m$-th element of the resulting type-$l$ vector of the tensor product between $\mathbf{x}^{(l_1)}$ and $\mathbf{x}^{(l_2)}$ is given by:

$$(\mathbf{x}^{(l_1)} \otimes_{cg} \mathbf{x}^{(l_2)})_m^{(l)} = \sum_{m_1=-l_1}^{l_1} \sum_{m_2=-l_2}^{l_2} C_{(l_1,m_1)(l_2,m_2)}^{(l,m)} x_{m_1}^{(l_1)} x_{m_2}^{(l_2)}, \quad (19)$$

where $C_{(l_1,m_1)(l_2,m_2)}^{(l,m)}$ are the Clebsch-Gordan coefficients. Interestingly, the Clebsch-Gordan coefficients are also closely related to angular momentum theory in quantum mechanics (Varshalovich et al., 1988), and have various intriguing properties that have underlying relationships to other important concepts used in this work.

**Properties of Clebsch-Gordan coefficients.** The Clebsch-Gordan coefficients have the following selection rules: $C_{(l_1,m_1)(l_2,m_2)}^{(l,m)} = 0$ if $l < |l_1 - l_2|$ or $l > l_1 + l_2$. That is, a type-0 vector and a type-1 vector cannot produce a type-2 vector via the Clebsch-Gordan tensor product. The Clebsch-Gordan coefficients have the following orthogonality and symmetry properties:

$$\text{Orthogonality: } \sum_{m_1,m_2} C_{(l_1,m_1)(l_2,m_2)}^{(l,m)} C_{(l_1,m_1)(l_2,m_2)}^{(l,m')} = \delta_{ll'}\delta_{mm'}$$

$$\sum_{l,m} C_{(l_1,m_1)(l_2,m_2)}^{(l,m)} C_{(l_1,m_1')(l_2,m_2')}^{(l,m)} = \delta_{m_1 m_1'}\delta_{m_2 m_2'} \quad (20)$$

$$\text{Symmetry: } C_{(l_1,m_1)(l_2,m_2)}^{(l,m)} = (-1)^{l_1+l_2-l} C_{(l_1,-m_1)(l_2,-m_2)}^{(l,-m)}$$

$$= (-1)^{l_1+l_2-l} C_{(l_2,m_2)(l_1,m_1)}^{(l,m)} \quad (21)$$

$$= (-1)^{l_1-m_1} \sqrt{\frac{2l+1}{2l_2+1}} C^{(l_2,-m_2)}_{(l_1,m_1)(l,-m)} = (-1)^{l_2+m_2} \sqrt{\frac{2l+1}{2l_1+1}} C^{(l_1,-m_1)}_{(l,-m)(l_2,m_2)}$$

$$= (-1)^{l_1-m_1} \sqrt{\frac{2l+1}{2l_2+1}} C^{(l_2,m_2)}_{(l,m)(l_1,-m_2)} = (-1)^{l_2+m_2} \sqrt{\frac{2l+1}{2l_2+1}} C^{(l_1,m_1)}_{(l_2,-m_2)(l,m)}$$

## A.6 RELATIONSHIP BETWEEN CLEBSCH-GORDAN, WIGNER 3-J AND GAUNT COEFFICIENTS.

We are motivated by the orthogonality and symmetry of the Clebsch-Gordan coefficients to explore further relations to two other concepts, Wigner 3-j symbols and Gaunt coefficients (Wigner, 1965). These concepts are widely used in quantum mechanics, and inspire the approaches presented in the main body of this work.

In quantum mechanics, the Wigner 3-j symbols are an alternative to Clebsch-Gordan coefficients, with the purpose of adding angular momenta. Mathematically, the Wigner 3-j symbols are related to Clebsch-Gordan coefficients as:

$$C^{(l,m)}_{(l_1,m_1)(l_2,m_2)} = \frac{(-1)^{-l_1+l_2-m}}{\sqrt{2l+1}} \begin{pmatrix} ,l_1 & l_2 & l \\ m_1 & m_2 & -m \end{pmatrix} \tag{22}$$

where $\begin{pmatrix} l_1 & l_2 & l \\ m_1 & m_2 & m \end{pmatrix}$ denotes the Wigner 3-j symbol that has the following explicit expression:

$$\begin{pmatrix} l_1 & l_2 & l \\ m_1 & m_2 & m \end{pmatrix} \equiv \delta(m_1+m_2+m,0)(-1)^{l_1-l_2-m}$$

$$\times \sqrt{\frac{(l_1+l_2-l)!\,(l_1-l_2+l)!\,(-l_1+l_2+l)!}{(l_1+l_2+l+1)!}}$$

$$\times \sqrt{(l_1-m_1)!\,(l_1+m_1)!\,(l_2-m_2)!\,(l_2+m_2)!\,(l-m)!\,(l+m)!}$$

$$\times \sum_{k=K}^{N} \frac{(-1)^k}{k!\,(l_1+l_2-l-k)!\,(l_1-m_1-k)!\,(l_2+m_2-k)!\,(l-l_2+m_1+k)!\,(l-l_1-m_2+k)!}, \tag{23}$$

where $K$ and $N$ are set as $K = \max(0, l_2-l-m_1, l_1-l+m_2)$, $N = \min(l_1+l_2-l, l_1-m_1, l_2+m_2)$ and factorials of negative numbers are conventionally taken equal to zero. The Wigner 3-j symbols are highly symmetric. They are also invariant under an even permutation of their columns, and enumerate a phase factor under an odd permutation or changing the sign of the $m$ numbers. The Wigner 3-j symbols also have the so-called Regge symmetries (Zhedanov et al., 1988).

The Wigner 3-j symbols are related to spherical harmonics via the Gaunt coefficients. which are the integrals of the products of three spherical harmonics:

$$\int_0^{2\pi} \int_0^{\pi} Y^{(l_1)}_{m_1}(\theta,\psi) Y^{(l_2)}_{m_2}(\theta,\psi) Y^{(l)}_m(\theta,\psi) \sin\theta d\theta d\psi$$

$$= \sqrt{\frac{(2l_1+1)(2l_2+1)(2l+1)}{4\pi}} \begin{pmatrix} l_1 & l_2 & l \\ 0 & 0 & 0 \end{pmatrix} \begin{pmatrix} l_1 & l_2 & l \\ m_1 & m_2 & m \end{pmatrix}. \tag{24}$$

The Gaunt coefficients are used in quantum mechanical computations, e.g., multi-center molecular integrals (Huzinaga, 1967). In this work, we use their relations to the Clebsch-Gordan coefficients to accelerate the tensor product of representations, thereby improving the efficiency of different classes of equivariant operations. From the above relations, we conclude this subsection with the following equivalence:

$$\int_0^{2\pi} \int_0^{\pi} Y^{(l_1)}_{m_1}(\theta,\psi) Y^{(l_2)}_{m_2}(\theta,\psi) Y^{(l)}_m(\theta,\psi) \sin\theta d\theta d\psi = \tilde{C}^{(l)}_{(l_1,l_2)} C^{(l,m)}_{(l_1,m_1)(l_2,m_2)}, \tag{25}$$

where $\tilde{C}^{(l)}_{(l_1,l_2)}$ is a constant only determined by $l_1, l_2$, and $l$. This equivalence inspires the development of the Gaunt Tensor Product presented in the main body of this work.

We note one difference between employing Clebsch-Gordan coefficients and Gaunt coefficients in the tensor product computation, namely that the Gaunt coefficients impose more symmetry

constraints. For instance, the tensor product with Gaunt coefficients excludes pseudovectors from the output. However, we empirically observe that this does not hurt performance (See the sanity check experiments in Section 4). We also note that several existing models that achieved strong performance in real-world applications (Brandstetter et al., 2022; Batzner et al., 2022; Batatia et al., 2022b) that consider the parity of irreps do not use irreps of even parities, e.g., pseudovectors, in computing equivariant representations. In these cases, the tensor product with Clebsch-Gordan coefficients is equivalent to the tensor product with Gaunt coefficients.

### A.7 ANGULAR MOMENTUM, TENSOR OPERATORS AND WIGNER-ECKART THEOREM

In the previous subsection, we introduced the relations between the Clebsch-Gordan coefficients and the Gaunt coefficients, which serve as the key to motivate the new perspective on tensor products of irreps via Gaunt coefficients in Section 3.1. For completeness, we thoroughly elaborate on the necessary knowledge to derive this equivalence in this subsection. First, we leverage the lens of angular momentum in quantum mechanics to describe spherical harmonics and Clebsch-Gordan coefficients. We then elaborate on the concepts of tensor operators and the Wigner-Eckart theorem, which we use to relate Clebsch-Gordan coefficients and Gaunt coefficients, as introduced below.

#### A.7.1 PRELIMINARY

We introduce several mathematical concepts that are used in quantum mechanics.

**Hilbert space and Dirac notation ("bra-ket" notation).** A *Hilbert space* $\mathcal{H}$ is an abstract vector space satisfying the following conditions: (1) it is equipped with the inner product operation; (2) it is a complete metric space with respect to the distance function induced by the inner product. A typical example of a Hilbert space is the three-dimensional Euclidean vector space $\mathbb{R}^3$, which is equipped with an inner product operation known as the dot product $\cdot : \mathbb{R}^3 \times \mathbb{R}^3 \to \mathbb{R}$. As a general and abstract concept, Hilbert spaces allow mathematical tools like linear algebra for (finite-dimensional) Euclidean vector spaces to spaces that may be infinite-dimensional, e.g., function spaces frequently used in physics.

In Hilbert space $\mathcal{H}$, we instead use *bra-ket notation* to describe our system. Vectors in the Hilbert space $\mathcal{H}$ are written as *kets*: $|\alpha\rangle \in \mathcal{H}$. The operational rules from Euclidean vector space are also generalized as,

1. Addition: $|\alpha\rangle + |\beta\rangle = |\beta\rangle + |\alpha\rangle$ (commutative), $|\alpha\rangle + (|\beta\rangle + |\gamma\rangle) = (|\alpha\rangle + |\beta\rangle) + |\gamma\rangle$ (associative);

2. Scalar product: $(c_1 + c_2)|\alpha\rangle = c_1|\alpha\rangle + c_2|\alpha\rangle$, $c_1(c_2|\alpha\rangle) = (c_1 c_2)|\alpha\rangle$, $c(|\alpha\rangle + |\beta\rangle) = c|\alpha\rangle + c|\beta\rangle$, $1|\alpha\rangle = |\alpha\rangle$, where the scalars are complex $c \in \mathbb{C}$ (as we will deal exclusively with complex Hilbert spaces for quantum mechanics);

3. Null ket $|\emptyset\rangle$: $|\alpha\rangle + |\emptyset\rangle = |\alpha\rangle$, $0|\alpha\rangle = |\emptyset\rangle$;

4. Inverse: any ket $|\alpha\rangle$ has a corresponding inverse $|-\alpha\rangle$, and we have $|\alpha\rangle + |-\alpha\rangle = |\emptyset\rangle$, $|-\alpha\rangle = -|\alpha\rangle$. For an $N$-dimensional Hilbert space, there exists a set of $N$ kets which constitute a basis, and any ket can be expressed as a sum of the basis kets, i.e., $|\alpha\rangle = \sum_{i=1}^{N} \alpha_i |\lambda_i\rangle, \alpha_i \in \mathbb{C}$.

We further define the inner product in a Hilbert space, i.e., a map which takes two kets and returns a scalar (complex number): $(\cdot, \cdot) : \mathcal{H} \times \mathcal{H} \to \mathbb{C}$. With this definition, we now define the *bra* notation, which is written as a backwards-facing ket $\langle\alpha|$. We have the one-to-one correspondence from kets to bras and vice-versa:

1. $|\alpha\rangle \leftrightarrow \langle\alpha|$;

2. $c_\alpha|\alpha\rangle + c_\beta|\beta\rangle \leftrightarrow c_\alpha^*\langle\alpha| + c_\beta^*\langle\beta|$,

where $c_\alpha^*, c_\beta^* \in \mathbb{C}$ are the complex conjugates of $c_\alpha, c_\beta \in \mathbb{C}$. By using the bra-ket notation, the inner product of two kets in a Hilbert space can be written as a product of a bra and a ket: $(|\alpha\rangle, |\beta\rangle) \equiv \langle\alpha|\beta\rangle$. In this notation, the inner product has the following properties:

1. linearity (in the bras): $(c\langle\alpha|)|\beta\rangle = c\langle\alpha|\beta\rangle$ and $((\langle\alpha_1| + \langle\alpha_2|)|\beta\rangle = \langle\alpha_1|\beta\rangle + \langle\alpha_2|\beta\rangle$;

2. linearity (in the kets): $\langle\alpha|(c|\beta\rangle) = c\langle\alpha|\beta\rangle$ and $\langle\alpha|(|\beta_1\rangle + |\beta_2\rangle) = \langle\alpha|\beta_1\rangle + \langle\alpha|\beta_2\rangle$;

3. conjugate symmetry: $\langle\alpha|\beta\rangle = \langle\beta|\alpha\rangle^*$;

4. positive semi-definiteness: $\langle\alpha|\alpha\rangle \geq 0$.

From (3), we can see that the order of the kets matters in the product. In fact, a bra denotes a linear map that maps kets in $\mathcal{H}$ to $\mathbb{C}$, i.e., $(|\alpha\rangle, \cdot) \equiv \langle\alpha|$. We say that two kets $|\alpha\rangle$ and $|\beta\rangle$ are orthogonal if their inner product is zero, i.e., $\langle\alpha|\beta\rangle = \langle\beta|\alpha\rangle = 0 \Rightarrow |\alpha\rangle \perp |\beta\rangle$. Again, analogous to Euclidean vector space, we use $||\alpha|| \equiv \sqrt{\langle\alpha|\alpha\rangle}$ to denote the norm of the ket $|\alpha\rangle$.

As previously mentioned, the three-dimensional Euclidean vector space is one example of the Hilbert space with finite dimensions. Here, we provide an example of infinite-dimensional Hilbert spaces for illustration. Consider a function $y(x) = \sum_{n=1}^{\infty} c_n \sin(\frac{n\pi x}{L}), x \in \mathbb{R}$: each ket in this space represents a function, and the basis kets are given by the infinite set of functions $|n\rangle = \sin(\frac{n\pi x}{L})$ (we can use $n$ in the ket notation for brevity, because it is the identity index in this Hilbert space). In this space, the inner product is defined between functions as $\langle y_1|y_2\rangle \equiv \frac{2}{L} \int y_1(x)y_2(x)\mathrm{d}x$.

**Operators.** To describe the mapping between kets in a Hilbert space, we introduce the concept of *operators*, i.e., $\hat{O} : \mathcal{H} \to \mathcal{H}$. In general, properties of operators include:

1. Equality: $\hat{A} = \hat{B}$ if $\hat{A}|\alpha\rangle = \hat{B}|\alpha\rangle$ for all $|\alpha\rangle$;

2. Null: there exists a special operator $\hat{0}$ such that $\hat{0}|\alpha\rangle = |\emptyset\rangle$ for all $|\alpha\rangle$;

3. Identity: there exists another special operator $\hat{1}$ such that $\hat{1}|\alpha\rangle = |\alpha\rangle$ for all $|\alpha\rangle$;

4. Addition: $\hat{A} + \hat{B} = \hat{B} + \hat{A}$ and $\hat{A} + (\hat{B} + \hat{C}) = (\hat{A} + \hat{B}) + \hat{C}$;

5. Scalar multiplication: $(c\hat{A})|\alpha\rangle = \hat{A}(c|\alpha\rangle)$;

6. Multiplication: $(\hat{A}\hat{B})|\alpha\rangle = \hat{A}(\hat{B}|\alpha\rangle)$.

A linear operator $\hat{A}$ has the distributive law, i.e., $\hat{A}(c_\alpha|\alpha\rangle + c_\beta|\beta\rangle) = c_\alpha\hat{A}|\alpha\rangle + c_\beta\hat{A}|\beta\rangle$. Using $\mathbb{R}^3$ as an example, linear operators are thus matrices $\mathbb{R}^{3\times3}$.

We use $\hat{A}^{-1}$ to denote the inverse operator of $\hat{A}$, and $\hat{A}^{-1}\hat{A}|\alpha\rangle = \hat{1}|\alpha\rangle$. Equipped with the inner product of a Hilbert space, we can define the *adjoint* of an operator denoted with a dagger by $(|\alpha\rangle, \hat{A}|\beta\rangle) = (\hat{A}^\dagger|\alpha\rangle, |\beta\rangle)$ for any choice of $|\alpha\rangle$ and $|\beta\rangle$. Combined with the operations, the adjoint has the following properties: (1) $(c\hat{A})^\dagger = c^*\hat{A}^\dagger$; (2) $(\hat{A} + \hat{B})^\dagger = \hat{A}^\dagger + \hat{B}^\dagger$; (3) $(\hat{A}\hat{B})^\dagger = \hat{B}^\dagger\hat{A}^\dagger$. One important class of operators are those that are self-adjoint, i.e., $\hat{A}^\dagger = \hat{A}$, which are also known as *Hermitian operators*. If the adjoint gives the inverse operator, i.e., $\hat{U}^\dagger = \hat{U}^{-1}$, these are called *unitary operators*, and act like a rotation in the Hilbert space.

Similar to matrices, the order of the operators matters when we multiply them, and the difference between the two orderings has a special symbol called the *commutator*: $[\hat{A}, \hat{B}] = \hat{A}\hat{B} - \hat{B}\hat{A}$. If the commutator of two operators is zero, we say they commute. Properties of commutators include:

1. $[\hat{A}, \hat{A}] = 0$;

2. $[\hat{A}, \hat{B}] = -[\hat{B}, \hat{A}]$;

3. $[\hat{A} + \hat{B}, \hat{C}] = [\hat{A}, \hat{C}] + [\hat{B}, \hat{C}]$;

4. $[\hat{A}, \hat{B}\hat{C}] = [\hat{A}, \hat{B}]\hat{C} + \hat{B}[\hat{A}, \hat{C}]$;

5. $[\hat{A}\hat{B}, \hat{C}] = \hat{A}[\hat{B}, \hat{C}] + [\hat{A}, \hat{C}]\hat{B}$;

6. $[\hat{A}, [\hat{B}, \hat{C}]] + [\hat{B}, [\hat{C}, \hat{A}]] + [\hat{C}, [\hat{A}, \hat{B}]] = 0$.

One intriguing property of two Hermitian operators that commute with each other is that we can simultaneously find their eigenkets. This property relates to the measurement in quantum mechanics.

**Eigenkets, basis kets and matrix elements of operators.** Analogous to linear algebra in the Euclidean vector space, we define the *eigenkets* of an operator $\hat{A}$ as the kets which are left invariant up to a scalar multiplication, i.e., $\hat{A}|a\rangle = a|a\rangle$, where the scalar $a$ is the eigenvalue associated with eigenket $|a\rangle$ (by convention, we use eigenvalues as indexes for eigenkets). Note that eigenkets are also called *eigenstates* in the context of quantum mechanics.

Recall that each ket in a Hilbert space can be expressed as a linear combination of the *basis kets*. In fact, there always exists an orthonormal basis, i.e., a set of basis kets which are mutually orthogonal and have norm 1. Let us use $|e_n\rangle$ to denote such an orthonormal basis. Then we have $\langle e_m|\alpha\rangle = \sum_n \alpha_n \langle e_m|e_n\rangle = \alpha_m$, and $|\alpha\rangle = \sum_n (|e_n\rangle\langle e_n|)|\alpha\rangle$. In fact, we introduce a special operator called *projection operator*, i.e., $\Lambda_n = |e_n\rangle\langle e_n|$, which project $|\alpha\rangle$ onto one of the basis kets $|e_n\rangle$.

By using the basis kets, we can now represent any ket as a column vector consisting of the coefficients corresponding to the basis kets: $|\psi\rangle = \sum_n \psi_n |e_n\rangle = [\psi_1, ..., \psi_n, ...]^\top$, and the associated bra $\langle\psi|$ is the conjugate transpose, i.e., a row vector whose entries are the complex conjugates of the $\psi_n$'s: $\langle\psi| = \sum_n \psi_n^* \langle e_n| = [\psi_1^*, ..., \psi_n^*, ...]$. Now, we have $\langle\chi|\psi\rangle = \sum_i \chi_i^* \psi_i$ and $(|\psi\rangle\langle\chi|)_{ij} = \psi_i \chi_j^*$. Specially, we have $|e_1\rangle\langle e_1|$ being an almost zero matrix except for the $(1,1)$-th entry being 1.

Thus, any operator $\hat{A}$ can be similarly represented as a matrix form:

$$\hat{A} = \sum_m \sum_n |e_m\rangle\langle e_m|\hat{A}|e_n\rangle\langle e_n| \rightarrow \begin{pmatrix} \langle e_1|\hat{A}|e_1\rangle & \langle e_1|\hat{A}|e_2\rangle & ... \\ \langle e_2|\hat{A}|e_1\rangle & \langle e_2|\hat{A}|e_2\rangle & ... \\ ... & ... & ... \end{pmatrix} \tag{26}$$

The entries of the matrix $\langle e_m|\hat{A}|e_n\rangle$ are known as *matrix elements* of operators. There also exists a relation between the matrix elements of $\hat{A}$ and its adjoint $\hat{A}^\dagger$: $\langle e_m|\hat{A}|e_n\rangle = \langle e_n|\hat{A}^\dagger|e_m\rangle^*$.

### A.7.2 A Brief Overview of Angular Momentum in Quantum Mechanics

**Momentum Operator and Position Operator.** Before we go into the concept of angular momentum in quantum mechanics, we discuss two basic examples of operators that we will use later. Here, we use a concrete example to see how previously introduced concepts are reflected in real-world systems: the 1D De Broglie plane wave function for a free particle with momentum $p$ and energy $E$, i.e., $\Psi(x,t) = e^{i(kx-\omega t)}$, where $p = \hbar k, E = \hbar\omega, E = \frac{p^2}{2m}$, $\hbar$ denotes the Planck Constant, and $m$ denotes the mass of the particle. To extract the value of momentum from the wave function, we can use the following operator:

$$\frac{\hbar}{i}\frac{\partial}{\partial x}\Psi(x,t) = \frac{\hbar}{i}\frac{\partial}{\partial x}e^{i(kx-\omega t)} = \frac{\hbar}{i}(ik)e^{i(kx-\omega t)} = \frac{\hbar}{i}(ik)\Psi(x,t) = \hbar k\Psi(x,t) = p\Psi(x,t)$$
$$\tag{27}$$

We then define the *momentum operator*, $\hat{p}$, as $\hat{p} = \frac{\hbar}{i}\frac{\partial}{\partial x} = -i\hbar\frac{\partial}{\partial x}$. Moreover, we can see that $\Psi(x,t)$ indeed lies in a Hilbert functional space, and is an eigenstate of the momentum operator $\hat{p}$ because $\Psi(x,t)$ is left invariant up to a scalar $p$ being multiplied after the action of $\hat{p}$.

Another important operator is the *position operator* $\hat{x}$, which acts on functions of $x$, and gives another function of $x$ as follows:

$$\hat{x}f(x) \equiv xf(x). \tag{28}$$

In fact, both the operator $\hat{x}$ and $\hat{p}$ are related. Given some arbitrary function $\phi(x)$, the commutator $[\hat{x}, \hat{p}]$ has the following equivalence:

$$[\hat{x}, \hat{p}]\phi(x) = (\hat{x}\hat{p} - \hat{p}\hat{x})\phi(x) = \hat{x}\hat{p}\phi(x) - \hat{p}\hat{x}\phi(x) = \hat{x}(\hat{p}\phi(x)) - \hat{p}(\hat{x}\phi(x)) = \hat{x}\left(-i\hbar\frac{\partial\phi(x)}{\partial x}\right) - \hat{p}(x\phi(x)),$$

$$= -i\hbar x\frac{\partial\phi(x)}{\partial x} + i\hbar\frac{\partial}{\partial x}(x\phi(x)) = -i\hbar x\frac{\partial\phi(x)}{\partial x} + i\hbar x\frac{\partial\phi(x)}{\partial x} + i\hbar\phi(x) = i\hbar\phi(x). \tag{29}$$

Thus, we obtain a fundamental commutation relation in quantum mechanics: $[\hat{x}, \hat{p}] = i\hbar$. It is also straightforward to generalize both $\hat{x}$ and $\hat{p}$ to three dimensions:

$$\hat{\mathbf{p}} = (\hat{p}_x, \hat{p}_y, \hat{p}_z) = -i\hbar\vec{\nabla} = -i\hbar\left(\frac{\partial}{\partial x}, \frac{\partial}{\partial y}, \frac{\partial}{\partial z}\right),$$
$$\hat{\mathbf{x}} = (\hat{x}, \hat{y}, \hat{z}). \tag{30}$$

And the commutation relationships are as follows:

$$[\hat{x}, \hat{p}_x] = i\hbar, [\hat{y}, \hat{p}_y] = i\hbar, [\hat{z}, \hat{p}_z] = i\hbar, \tag{31}$$

with all other commutators involving the three coordinates and momenta being zero.

**Angular Momentum Operator.** We are now ready to introduce the concept of angular momentum in quantum mechanics. This helps us understand the underlying relations between spherical harmonics and Clebsch-Gordan coefficients, as stated in Section 3.1. In classical physics, the angular momentum of a particle with momentum $\mathbf{p}$ and position $\mathbf{r}$ is defined by $\mathbf{L} = \mathbf{r} \times \mathbf{p}$, and each component of $\mathbf{L} = (L_x, L_y, L_z)$ is given by $L_x = yp_z - zp_y, L_y = zp_x - xp_z, L_z = xp_y - yp_x$. Similarly, the *angular momentum operator* $\hat{\mathbf{L}} = (\hat{L}_x, \hat{L}_y, \hat{L}_z)$ can be obtained as follows:

$$\begin{aligned} \hat{L}_x &= \hat{y}\hat{p}_z - \hat{z}\hat{p}_y, \\ \hat{L}_y &= \hat{z}\hat{p}_x - \hat{x}\hat{p}_z, \\ \hat{L}_z &= \hat{x}\hat{p}_y - \hat{y}\hat{p}_x. \end{aligned} \tag{32}$$

From the definition, we can check that the angular momentum components are Hermitian operators: $\hat{L}_x^\dagger = \hat{L}_x, \hat{L}_y^\dagger = \hat{L}_y, \hat{L}_z^\dagger = \hat{L}_z$. The square of $\hat{\mathbf{L}}$ is also Hermitian, which is $\hat{\mathbf{L}}^2 = \hat{L}_x^2 + \hat{L}_y^2 + \hat{L}_z^2$. Therefore, all the angular momentum operators have real eigenvalues.

**Eigenstates and eigenvalues of Angular Momentum Operators.** Recall that we can simultaneously find the eigenstates of two Hermitian operators that commute with each other. In fact, we have the following commutation relations for angular momentum operators:

$$[\hat{L}_x, \hat{L}_y] = i\hbar\hat{L}_z, [\hat{L}_y, \hat{L}_z] = i\hbar\hat{L}_x, [\hat{L}_z, \hat{L}_x] = i\hbar\hat{L}_y, \tag{33}$$

$$[\hat{L}_x, \hat{\mathbf{L}}^2] = 0, [\hat{L}_x, \hat{\mathbf{L}}^2] = 0, [\hat{L}_x, \hat{\mathbf{L}}^2] = 0. \tag{34}$$

This means that we can simultaneously find the eigenstates of $\hat{L}_z$ and $\hat{\mathbf{L}}^2$ (also for $\hat{L}_x, \hat{L}_y$). To achieve this, let us first turn our view from the Cartesian coordinate systems to the spherical coordinate system, for convenience:

$$\begin{aligned} x &= r\sin\theta\cos\phi, \quad r = \sqrt{x^2 + y^2 + z^2}, \\ y &= r\sin\theta\sin\phi, \quad \theta = \cos^{-1}\left(\frac{z}{r}\right), \\ z &= r\cos\theta, \quad \phi = \tan^{-1}\left(\frac{y}{x}\right). \end{aligned} \tag{35}$$

By the definition $\hat{L}_z = \hat{x}\hat{p}_y - \hat{y}\hat{p}_x$, we have:

$$\hat{L}_z = -i\hbar\left(x\frac{\partial}{\partial y} - y\frac{\partial}{\partial x}\right). \tag{36}$$

In fact, this is related to $\frac{\partial}{\partial\phi}$ by the chain rule: $\frac{\partial}{\partial\phi} = \frac{\partial x}{\partial\phi}\frac{\partial}{\partial x} + \frac{\partial y}{\partial\phi}\frac{\partial}{\partial y} + \frac{\partial z}{\partial\phi}\frac{\partial}{\partial z} = x\frac{\partial}{\partial y} - y\frac{\partial}{\partial x}$, i.e.,

$$\begin{aligned} \hat{L}_z &= -i\hbar\frac{\partial}{\partial\phi}, \\ \hat{L}_x &= i\hbar\left(\sin\phi\frac{\partial}{\partial\theta} + \cot\theta\cos\phi\frac{\partial}{\partial\phi}\right), \\ \hat{L}_y &= i\hbar\left(-\cos\phi\frac{\partial}{\partial\theta} + \cot\theta\sin\phi\frac{\partial}{\partial\phi}\right), \\ \hat{\mathbf{L}}^2 &= -\hbar^2\left[\frac{1}{\sin\theta}\frac{\partial}{\partial\theta}\left(\sin\theta\frac{\partial}{\partial\theta}\right) + \frac{1}{\sin^2\theta}\frac{\partial^2}{\partial\phi^2}\right]. \end{aligned} \tag{37}$$

Since we can construct the simultaneous eigenstates of both $\hat{L}_z$ and $\hat{\mathbf{L}}^2$, we use the union of their eigenvalues $(l, m)$ to index these simultaneous eigenstates as $|l, m\rangle$, and establish the following equations:

$$\begin{aligned} \hat{L}_z|l, m\rangle &= m|l, m\rangle, \\ \hat{\mathbf{L}}^2|l, m\rangle &= l|l, m\rangle, \end{aligned} \tag{38}$$

where $m, l \in \mathbb{R}$ because both $\hat{L}_z$ and $\hat{\mathbf{L}}^2$ are Hermitian operators. Let us now solve the first eigenvalue equation using the coordinate system for $\hat{L}_z = -i\hbar \frac{\partial}{\partial \phi}$, and denote $|l, m\rangle$ using its functional form $\psi_{l,m}$: $-i\hbar \frac{\partial \psi_{l,m}}{\partial \phi} = m \psi_{l,m} \rightarrow \frac{\partial \psi_{l,m}}{\partial \phi} = i \frac{m}{\hbar} \psi_{l,m}$. From this equation, we can determine the $\phi$ dependence of the solution and rewrite $\psi_{l,m}(\theta, \phi) = e^{i \frac{m}{\hbar} \phi} P_{l,m}(\theta)$, where $P_{l,m}(\theta)$ captures the still undetermined $\theta$ dependence of the eigenstate $\psi_{l,m}$. As a function of the angles, it is required that $\psi_{l,m}$ is uniquely defined: $\psi_{l,m}(\theta, \phi + 2\pi) = \psi_{l,m}(\theta, \phi)$, from which we can obtain $e^{i \frac{m}{\hbar} 2\pi} = 1$. If we instead write this eigenvalue equation as $\hat{L}_z |l, m\rangle = \hbar m |l, m\rangle$, we thus have the fact that $m \in \mathbb{Z}$. Similarly, it can be shown that the possible values of $l$ are $l \in \mathbb{N}$ and $-l \leq m \leq l$, with the eigenvalue equations being written as,

$$\hat{L}_z |l, m\rangle = \hbar m |l, m\rangle,$$
$$\hat{\mathbf{L}}^2 |l, m\rangle = \hbar^2 l(l+1)|l, m\rangle. \tag{39}$$

After determining the eigenvalues of operators $\hat{L}_z$ and $\hat{\mathbf{L}}^2$, the next step is to obtain the form of the eigenstates $|l, m\rangle$. By using the functional form, we rewrite the second equation as:

$$\hbar^2 \left[ \frac{1}{\sin\theta} \frac{\partial}{\partial \theta} \left( \sin\theta \frac{\partial}{\partial \theta} \right) + \frac{1}{\sin^2\theta} \frac{\partial^2}{\partial \phi^2} + l(l+1) \right] \psi_{l,m}(\theta, \phi) = 0. \tag{40}$$

This is exactly the *Laplacian equation* in the spherical domain (Sommerfeld, 1949), and the solution is exactly the spherical harmonics $\psi_{l,m}(\theta, \phi) \equiv Y_m^{(l)}(\theta, \phi)$ with $-l \leq m \leq l, l = 0, 1, 2, ...,$, as introduced in Section A.4. In the context of quantum mechanics, it is conventional to use the ket $|l, m\rangle$ to denote the spherical harmonics $Y_m^{(l)}$, and specify it in the spherical coordinate representation as $\langle \theta, \phi | l, m \rangle \equiv Y_m^{(l)}(\theta, \phi)$. In fact, the set of kets $\{|l, m\rangle | -l \leq m \leq l\}$ exactly forms an orthonormal basis for the Hilbert space with $2l + 1$ dimension.

**Addition of Angular Momentum Operators.** In quantum mechanics, we often consider systems with two physically different angular momenta, e.g., the orbital angular momenta of two electrons. It means that the angular momentum operators jointly act on two Hilbert spaces with different dimensions, which prompts us to define the concept of the *total angular momentum operator*. Given two angular momentum operators $\hat{j}_1$ and $\hat{j}_2$ acting on spaces $V_1$ and $V_2$ with $2l_1 + 1$ and $2l_2 + 1$ dimensions respectively, these two spaces are spanned by the eigenstates $|l_1, m_1\rangle, -l_1 \leq m_1 \leq l_1$ and $|l_2, m_2\rangle, -l_2 \leq m_2 \leq l_2$. The total angular momentum operator is the addition of these two operators, and acts on the tensor product space $V_3 \equiv V_1 \otimes V_2$ with $(2l_1 + 1) \times (2l_2 + 1)$ dimension, which is spanned by $|l_1, m_1, l_2, m_2\rangle \equiv |l_1, m_1\rangle \otimes |l_2, m_2\rangle, -l_1 \leq m_1 \leq l_1, -l_2 \leq m_2 \leq l_2$. We call $|l_1, m_1, l_2, m_2\rangle$ the *uncoupled tensor product basis*. Given any angular momentum operator $\hat{j}$, it is defined to act on states in $V_3$ in the following manner:

$$(\hat{j} \otimes \hat{\mathbf{1}})|l_1, m_1, l_2, m_2\rangle \equiv \hat{j}|l_1, m_1\rangle \otimes |l_2, m_2\rangle,$$
$$(\hat{\mathbf{1}} \otimes \hat{j})|l_1, m_1, l_2, m_2\rangle \equiv |l_1, m_1\rangle \otimes \hat{j}|l_2, m_2\rangle, \tag{41}$$

where $\hat{\mathbf{1}}$ denotes the identity operator. The total angular momentum operators are then defined as:

$$\hat{\mathbf{J}} \equiv \hat{j}_1 \otimes \hat{\mathbf{1}} + \hat{\mathbf{1}} \otimes \hat{j}_2 \tag{42}$$

Similar to Eqn. (33), the total angular momentum operators also have the following commutative relationship:

$$[\hat{J}_x, \hat{J}_y] = i\hbar \hat{J}_z, [\hat{J}_y, \hat{J}_z] = i\hbar \hat{J}_x, [\hat{J}_z, \hat{J}_x] = i\hbar \hat{J}_y, \tag{43}$$

$$[\hat{J}_x, \hat{\mathbf{J}}^2] = 0, [\hat{J}_x, \hat{\mathbf{J}}^2] = 0, [\hat{J}_x, \hat{\mathbf{J}}^2] = 0 \tag{44}$$

Therefore, a set of *coupled eigenstates* also simultaneously exists for $\hat{\mathbf{J}}^2$ and $\hat{J}_z$, which we denote as $|l_1, l_2; L, M\rangle$. This has the following eigenvalue equations:

$$\hat{J}_z |l_1, l_2; L, M\rangle = \hbar M |l_1, l_2; L, M\rangle$$
$$\hat{\mathbf{J}}^2 |l_1, l_2; L, M\rangle = \hbar^2 L(L+1)|l_1, l_2; L, M\rangle \tag{45}$$

where $-L \leq M \leq L, |l_1 - l_2| \leq L \leq l_1 + l_2$, and the total number of eigenstates of the total angular momentum operator is exactly equal to the dimension of the tensor product space $V_3$:

$\sum_{L=|l_1-l_2|}^{l_1+l_2}(2L+1) = (2l_1+1) \times (2l_2+1)$. Both the uncoupled tensor product basis and the coupled eigenstates form a basis for the tensor product space, i.e., they are related via a change of basis:

$$|l_1, l_2; L, M\rangle = \sum_{m_1=-l_1}^{l_1} \sum_{m_2=-l_2}^{l_2} |l_1, m_1, l_2, m_2\rangle\langle l_1, m_1, l_2, m_2|L, M\rangle, \tag{46}$$

where the expansion coefficients $\langle l_1, m_1, l_2, m_2|L, M\rangle$ are the Clebsch-Gordan coefficients denoted as $C^{(L,M)}_{(l_1,m_1)(l_2,m_2)}$, which we previously introduced in Appendix A.5.

In summary, spherical harmonics correspond to the eigenstates of angular momentum operators, while the Clebsch-Gordan coefficients serve as the expansion coefficients of coupled eigenstates of total angular momentum operators in an uncoupled tensor product basis. This indicates an underlying relationship between these two concepts, which we will further reveal by using the Wigner-Eckart theorem below.

### A.7.3 TENSOR OPERATORS AND WIGNER-ECKART THEOREM

**Rotation Operator.** As we introduced in Section 2, representations of the rotation group $SO(3)$ on $\mathbb{R}^3$ are parameterized by orthogonal transformation matrices $\mathbf{R} \in \mathbb{R}^{3\times3}, \det(\mathbf{R}) = 1$, which are applied to $\mathbf{x} \in \mathbb{R}^3$ via matrix multiplication $\mathbf{R}\mathbf{x}$. Each rotation matrix $\mathbf{R}$ has the explicit form that is specified by the rotation axis and angle, e.g., rotation by angle $\phi$ about the $z$-axis is given by $\mathbf{R}_z(\phi) = \begin{pmatrix} \cos\phi & -\sin\phi & 0 \\ \sin\phi & \cos\phi & 0 \\ 0 & 0 & 1 \end{pmatrix}$. We also use $R(\vec{n}, \phi)$ to denote the rotation $g \in SO(3)$ that performs rotation by angle $\phi$ about the $\vec{n} \in \mathbb{R}^3, ||\vec{n}|| = 1$ direction. In quantum mechanics, given any rotation $R(\vec{n}, \phi)$, there exists a unitary operator $\hat{U}(\vec{n}, \phi)$ that transforms a ket from the unrotated coordinate system to the rotated one, i.e., $|\psi\rangle_R = \hat{U}(\vec{n}, \phi)|\psi\rangle$. The previously introduced angular momentum operator $\hat{\mathbf{L}}$ is exactly the generator of the *rotation operator* $\hat{U}$:

$$\hat{U}(\vec{n}, \phi) = \exp(\frac{-i(\hat{\mathbf{L}} \cdot \vec{n})\phi}{\hbar}). \tag{47}$$

Recall that given $l \in \mathbb{N}$, the eigenstates of angular momentum operators $|l, m\rangle, -l \le m \le l$ form a basis for the Hilbert space with $2l + 1$ dimension, i.e., any ket $|\alpha\rangle$ in this space can be expressed as $|\alpha\rangle = \sum_{m=-l}^{l} \alpha_m|l, m\rangle$. We can also calculate the matrix element of the rotation operator $\hat{U}(\vec{n}, \phi)$ corresponding to $g \in SO(3)$ by using this basis:

$$\mathbf{D}^{(l)}_{m'm}(g) \equiv \langle l, m'|\exp\left(\frac{-i(\hat{\mathbf{L}} \cdot \vec{n})\phi}{\hbar}\right)|l, m\rangle, \tag{48}$$

which is exactly the Wigner-D matrix in the shape of $(2l + 1) \times (2l + 1)$, as introduced in Section 2. Hence, the rotated ket $|\alpha'\rangle$ can be expressed as:

$$|\alpha'\rangle = \hat{U}(\vec{n}, \phi)|\alpha\rangle = \sum_{m',m} \alpha_m|l, m'\rangle\langle l, m'|\hat{U}(\vec{n}, \phi)|l, m\rangle = \sum_{m',m} \mathbf{D}^{(l)}_{m'm}(g)\alpha_m|m'\rangle. \tag{49}$$

Therefore, the rotation transformation for kets in the $2l+1$-dimensional space spanned by $|l, m\rangle, -l \le m \le l$ can also be computed through matrix multiplication:

$$\alpha'_{m'} = \sum_m \mathbf{D}^{(l)}_{m'm}(g)\alpha_m. \tag{50}$$

**Vector Operators and Tensor Operators.** Classically, a (three-dimensional) vector is defined by its properties under rotation: the three components corresponding to the Cartesian $x, y, z$ axes transform as $V_i \to \sum_{j=1}^{3} \mathbf{R}_{ij}V_j$. A tensor is a generalization of such a vector to an object with more than one index, e.g., $T_{ij}$ or $T_{ijk}$ (with $3 \times 3$ and $3 \times 3 \times 3$ components respectively), and satisfies the condition that these components mix among themselves under rotation by each individual index following the vector rule $T_{ijk} \to \sum \mathbf{R}_{il}\mathbf{R}_{jm}\mathbf{R}_{kn}T_{lmn}$. Tensors written like this are called

Cartesian tensors, and the number of indexes is denoted as the rank of the Cartesian tensor, i.e., a rank $n$ Cartesian tensor has $3^n$ components in total.

In quantum mechanics, an operator $\hat{V}$ is called a *vector operator* if the expected values of its three components in any ket transform like the components of a classical 3D vector under rotation:

$$\hat{U}^\dagger \hat{V}_i \hat{U} = \sum_j \mathbf{R}_{ij} \hat{V}_j, \tag{51}$$

where $\hat{U}$ denotes the corresponding unitary rotation operator of the rotation matrix $\mathbf{R}$. Given any ket $|\alpha\rangle$ and its rotated ket $|\alpha'\rangle$ via the rotation operator $\hat{U}$, we can check the expected value of the vector operator $\hat{V}$ in the rotated system:

$$\langle \alpha' | \hat{V}_i | \alpha' \rangle = \langle \alpha | \hat{U}^\dagger \hat{V}_i \hat{U} | \alpha \rangle = \sum_j \mathbf{R}_{ij} \langle \alpha | \hat{V}_j | \alpha \rangle. \tag{52}$$

This exactly meets the requirement for the expected values of its three components under rotation. Similarly, we can further generalize to *Cartesian tensor operators*, which satisfy the condition that the expected values of its components in any ket transform like the components of a classical tensor under rotation.

Although a Cartesian tensor is easy to denote, it is usually hard to work with, especially when dealing with rotations. Instead, we further define the concept of *spherical tensor operators*. Given a basis ket $|l, m\rangle$ for the $2l + 1$-dimensional Hilbert space, it is transformed via a rotation $\hat{U}(\vec{n}, \phi)$ as:

$$\hat{U}(\vec{n}, \phi) |l, m\rangle = \sum_{m'} |l, m'\rangle \langle l, m' | \hat{U}(\vec{n}, \phi) |l, m\rangle = \sum_{m'} |l, m'\rangle \mathbf{D}^{(l)}_{m'm}(g), \tag{53}$$

where $g \in SO(3)$ corresponds to the rotation $\hat{U}(\vec{n}, \phi)$. From this equation, we can see that the rotation acts separately on different $2l + 1$-dimensional Hilbert spaces spanned by different sets of basis kets $|l, m\rangle, l \in \mathbb{N}$. By doing this, we can handle this rotation. Spherical tensor operators of rank $l$ are defined as families $\mathbf{T}^{(l)} = (T^{(l)}_{-l}, ..., T^{(l)}_l)^\top$ of $2l + 1$ operators $T^{(l)}_m$ that satisfy the following constraint:

$$\hat{U}^\dagger T^{(l)}_m \hat{U} = \sum_{m'=-l}^{l} T^{(l)}_{m'} \mathbf{D}^{(l)}_{m'm}(g). \tag{54}$$

Since the transformation rule of the spherical tensor operators originates from the properties of the eigenstates of angular momentum operator $|l, m\rangle$, spherical harmonics $Y^{(l)}_m$ are a special class of spherical tensor operators.

**Wigner-Eckart theorem.** Recall that all information about an operator $\hat{A} : \mathcal{H} \to \mathcal{H}$ is encoded by its matrix elements $\langle e_m | \hat{A} | e_n \rangle$ with the basis $e_m, m = 1, ...,$. From the definitions of vector operators and spherical harmonics operators, we can see their underlying symmetry constraints, indicating restricted freedom. Due to this, the matrix elements of such operators are fully specified by a comparatively small number of elements, which are called reduced matrix elements in the context of quantum mechanics. This reduction for operators is exactly described by the Wigner-Eckart theorem. For clarity, we discuss this theorem in its most popular form, i.e., for spherical tensor operators.

Given spherical tensor operators of rank $j$, it is natural to use the eigenstates of angular momentum operators $|l, m\rangle$ as the basis to calculate the matrix elements. By fixing $j, l, J$, there are $2j + 1$ components $T^j_m$ of the spherical tensor operator, $2l + 1$ basis kets $|l, m\rangle$ and $2J + 1$ basis bras $\langle J, M|$. Therefore, there are $(2j + 1) \times (2l + 1) \times (2J + 1)$ matrix elements $\langle J, M | T^{(j)}_m | l, n \rangle$. The following Wigner-Eckart theorem reveals how these matrix elements are specified.

**Theorem A.1** (Wigner-Eckart theorem for Spherical Tensor Operators, from Jeevanjee (2011)). *Let $j, l, J \in \mathbb{N}_{\geq 0}$ and let $\mathbf{T}^{(j)}$ be a spherical tensor operator of rank $j$. Then there is a unique complex number, the* reduced matrix element $\lambda \in \mathbb{C}$ *(often written $\langle J \| \mathbf{T}^{(j)} \| l \rangle \in \mathbb{C}$), that completely determines any of the* $(2J + 1) \times (2j + 1) \times (2l + 1)$ *matrix elements $\langle J, M | T^{(j)}_m | l, n \rangle$:*

$$\langle J, M | T^{(j)}_m | l, n \rangle = \lambda \cdot C^{(J,M)}_{(l,n)(j,m)}. \tag{55}$$

The Wigner-Eckart theorem describes how the matrix elements ($\langle J, M | T_m^{(j)} | l, n \rangle$) of the spherical tensor operator ($\boldsymbol{T}^{(j)}$) in the basis of angular momentum eigenstates ($\langle J, M |$ and $| l, n \rangle$) are decomposed into the Clebsch-Gordan coefficients ($C_{(l,n)(j,m)}^{(J,M)}$) and the *reduced matrix elements* ($\langle J \| \boldsymbol{T}^{(j)} \| l \rangle$). We can see that the $\langle J \| \boldsymbol{T}^{(j)} \| l \rangle$ values do not depend on $M, n, m$, which are thus called reduced matrix elements. The Wigner-Eckart theorem elegantly reveals the underlying connections between matrix elements, which is extremely useful for simplifying calculations in quantum mechanics.

Furthermore, if we instantiate $\boldsymbol{T}^{(j)}$ as the spherical harmonics $Y^{(j)}$ and apply the Wigner-Eckart theorem, we obtain the following relation $\langle J, M | Y_m^{(j)} | l, n \rangle = \langle J \| Y^{(j)} \| l \rangle \cdot C_{(l,n)(j,m)}^{(J,M)}$, which can be explicitly rewritten using the definition of inner products in the Hilbert functional space:

$$\int_0^{2\pi} \int_0^{\pi} Y_{m_1}^{(l_1)}(\theta, \psi) Y_{m_2}^{(l_2)}(\theta, \psi) Y_m^{(l)}(\theta, \psi) \sin \theta \mathrm{d}\theta \mathrm{d}\psi = \tilde{C}_{(l_1, l_2)}^{(l)} C_{(l_1, m_1)(l_2, m_2)}^{(l, m)}, \tag{56}$$

where $\tilde{C}_{(l_1, l_2)}^{(l)}$ is a real constant only determined by $l_1, l_2$ and $l$. From Eqn. (56), the Clebsch-Gordan coefficients are mathematically related to the integrals of products of three spherical harmonics, which are known as the Gaunt coefficients (Wigner, 2012), and are denoted as $G_{(l_1, m_1)(l_2, m_2)}^{(l, m)}$. This allows us to obtain the relationship presented in Appendix A.6.

## B ADVANCES IN EQUIVARIANT NETWORKS FOR THE EUCLIDEAN GROUP

One standard way to enforce equivariant operations is through group convolutions, which define convolution filters as functions on groups (Cohen & Welling, 2016; 2017; Coors et al., 2018). However, the integral in the group convolution becomes intractable when dealing with the continuous Euclidean groups, motivating the development of Lie-algebra-based operations (Finzi et al., 2020; Hutchinson et al., 2021). Finzi et al. (2020) proposed LieConv, which determines the group convolution via lifting (mapping the input in vector space to a group element), and discretization of the convolution integral via the PointConv trick. Following a similar idea, Hutchinson et al. (2021) developed LieTransformer, which employs the self-attention strategy to dynamically re-weight the convolutional kernel. Nevertheless, such methods still induce high computational costs due to discretization and sampling, and the non-linearity of equivariant features is restricted.

Another line of work leverages vector operations involving scalar ($\mathbb{R}$) and vector ($\mathbb{R}^3$) features to keep equivariance (Satorras et al., 2021; Jing et al., 2021; Schütt et al., 2021; Deng et al., 2021; Haghighatlari et al., 2022; Thölke & Fabritiis, 2022; Wang & Chodera, 2023; Chen et al., 2023). EGNN (Satorras et al., 2021) uses the scalar-vector product to update the vector features under the equivariance constraints. GVP (Jing et al., 2021), PaiNN (Schütt et al., 2021) and TorchMD-Net (Thölke & Fabritiis, 2022) further leverage vector products to enhance the learning of scalar features. Chen et al. (2023) recently proposed a unified framework called GeoMFormer, to simultaneously learn effective scalar and vector features. The empirical performance and applicability of these models are limited, due to bounded model capacity (Joshi et al., 2023).

Recently, tensor products of irreducible representations (irreps) have become the dominant equivariant operations for the Euclidean group (Thomas et al., 2018; Drautz, 2019; Fuchs et al., 2020; Unke et al., 2021; Brandstetter et al., 2022; Batatia et al., 2022a;b; Nigam et al., 2022; Batzner et al., 2022; Frank et al., 2022; Liao & Smidt, 2023; Musaelian et al., 2023; Passaro & Zitnick, 2023; Yu et al., 2023b; Gong et al., 2023; Liao et al., 2024). With a solid mathematical foundation from representation theory, such operations allow modeling equivariant mappings between any $2l+1$-dimensional irreps space via Clebsch-Gordan coefficients (Wigner, 2012), and are proven to have universal approximation capability (Dym & Maron, 2021). State-of-the-art performance has been achieved in various real-world applications via these operations (Ramakrishnan et al., 2014; Chmiela et al., 2017; Chanussot et al., 2021).

Various works use pair-wise tensor products to develop equivariant neural networks. Tensor Field Network (TFN) (Thomas et al., 2018) uses tensor products of irreps to build an $SE(3)$-equivariant convolution layer to handle data in the three-dimensional Euclidean space. Following TFN, Fuchs et al. (2020) developed the SE(3)-Transformer, which incorporates the attention mechanism to enhance the feature learning. NequIP (Batzner et al., 2022) further extended the tensor product operations to be $E(3)$-equivariant, and achieved strong performance on learning interatomic potentials. Allegro (Musaelian et al., 2023) further developed a strictly local equivariant network using the

iterated tensor products of irreps. SEGNN (Brandstetter et al., 2022) generalizes the node and edge features to include vectors or tensors, and incorporates geometric and physical information by using the tensor product operations. So3krates (Frank et al., 2022) developed a modified attention mechanism for modeling the non-local effects in molecular modeling. In particular, the atomic interaction layer in So3krates computes the full tensor products of features containing irreps of different degrees. Equiformer (Liao & Smidt, 2023) uses the tensor product operations to build a scalable equivariant Transformer architecture, testing this on the baselines of the large-scale OC20 dataset (Chanussot et al., 2021). This extended Graph Transformers (Ying et al., 2021a;b; Shi et al., 2022; Luo et al., 2023; Zhang et al., 2023a) to equivariant networks for the Euclidean group. More recently, eSCN (Passaro & Zitnick, 2023) presented an interesting observation on the sparsity of spherical harmonics filters and developed an equivariant network by using the SO(2) convolution instead, to scale the degree of irreps up. Following eSCN, EquiformerV2 (Liao et al., 2024) combined the Equiformer backbone with the eSCN convolution for scaling to higher-degree representations, which has state-of-the-art performance on the OC20 dataset.

Existing works have also developed operations involving multiple tensor products among equivariant features (Equivariant Many-body Interaction). Atomic Cluster Expansion (ACE) (Drautz, 2019; Dusson et al., 2022) developed a systematic framework for constructing many-body interaction features to encode the atomic environment, which involves multiple tensor products among atomic basis functions. MACE (Batatia et al., 2022b) combines the many-body interaction features with message-passing neural networks, which shows strong performance for learning machine learning force fields (Kovács et al., 2023). Nigam et al. (2022); Batatia et al. (2022a) provided a thorough analysis of the theoretical properties and design space of Equivariant Many-body Interaction operations.

Equivariant neural networks based on tensor products of irreps have been widely used in modeling 3D data across a variety of real-world applications, e.g., force field modeling (Chmiela et al., 2017; Chanussot et al., 2021), physical dynamics simulation (Brandstetter et al., 2022), point cloud learning (Wu et al., 2015; Uy et al., 2019), and Quantum Hamiltonian Prediction (Unke et al., 2021; Li et al., 2022; Khrabrov et al., 2022; Yu et al., 2023b;a). PhiSNet developed an SE(3)-equivariant message passing network by using the tensor product of irreps. In particular, to predict the Hamiltonian matrix elements, operations in the Equivariant Feature Interaction class are used for coupling atom features themselves (for diagonal block prediction), or features of atom pairs (for non-diagonal block prediction), which models interaction of irreps from different dimensional space and objects. Similarly, DeepH-E(3) (Li et al., 2022) further uses the tensor product operations to incorporate the spin-orbit coupling effects. QHNet (Yu et al., 2023b) simplifies the PhiSNet architecture by controlling the number of tensor product operations.

Finally, for comprehensive surveys on equivariant networks for the Euclidean group and their applications, we refer the interested readers to Bronstein et al. (2021); Han et al. (2022); Zhang et al. (2023b); Duval et al. (2023).

## C  IMPLEMENTATION DETAILS

In this section, we provide more details on designing efficient counterparts for operations in the Equivariant Feature Interaction, Equivariant Convolution, and Equivariant Many-body Interaction categories.

**Equivariant Feature Interactions.** Given two features $\tilde{\mathbf{x}}^{(L_1)}, \tilde{\mathbf{y}}^{(L_2)}$ containing irreps of up to degree $L_1$ and $L_2$ respectively, the operations in this class are commonly in the following form: $(\tilde{\mathbf{x}}^{(L_1)} \otimes_{\tilde{c}g}^{\mathbf{w}} \tilde{\mathbf{y}}^{(L_2)})^{(l)} = \sum_{l_1=0}^{L_1} \sum_{l_2=0}^{L_2} w_{l_1,l_2}^l (\mathbf{x}^{(l_1)} \otimes_{cg} \mathbf{y}^{(l_2)})^{(l)}$, where $w_{l_1,l_2}^l \in \mathbb{R}$ are the learnable weights for each $(l_1, l_2) \rightarrow l$ combination. By reparameterizing $w_{l_1,l_2}^l$ as $w_{l_1} \cdot w_{l_2} \cdot w_l$ where $w_{l_1}, w_{l_2}, w_l \in \mathbb{R}$, Eqn. (4) in the main body becomes:

$$(\tilde{\mathbf{x}}^{(L_1)} \otimes_{Ga\hat{u}nt}^{\mathbf{w}} \tilde{\mathbf{x}}^{(L_2)})_m^{(l)} = \sum_{l_1=0}^{L_1} \sum_{l_2=0}^{L_2} w_{l_1,l_2}^l (\mathbf{x}^{(l_1)} \otimes_{Gaunt} \mathbf{x}^{(l_2)})_m^{(l)}$$

$$= \sum_{l_1=0}^{L_1} \sum_{l_2=0}^{L_2} w_{l_1} w_{l_2} w_l (\mathbf{x}^{(l_1)} \otimes_{Gaunt} \mathbf{x}^{(l_2)})_m^{(l)}$$

$$= \sum_{l_1=0}^{L_1} \sum_{l_2=0}^{L_2} w_{l_1} w_{l_2} w_l \sum_{m_1=-l_1}^{l_1} \sum_{m_2=-l_2}^{l_2} G_{(l_1,m_1)(l_2,m_2)}^{(l,m)} x_{m_1}^{(l_1)} x_{m_2}^{(l_2)}$$

$$= \sum_{l_1=0}^{L_1} \sum_{l_2=0}^{L_2} w_{l_1} w_{l_2} w_l \sum_{m_1=-l_1}^{l_1} \sum_{m_2=-l_2}^{l_2} x_{m_1}^{(l_1)} x_{m_2}^{(l_2)} \int_0^{2\pi} \int_0^\pi Y_{m_1}^{(l_1)}(\theta,\psi) Y_{m_2}^{(l_2)}(\theta,\psi) Y_m^{(l)}(\theta,\psi) \sin\theta \mathrm{d}\theta \mathrm{d}\psi$$

$$= w_l \int_0^{2\pi} \int_0^\pi (\sum_{l_1=0}^{L_1} \sum_{m_1=-l_1}^{l_1} w_{l_1} x_{m_1}^{(l_1)} Y_{m_1}^{(l_1)}(\theta,\psi))(\sum_{l_2=0}^{L_2} \sum_{m_2=-l_2}^{l_2} w_{l_2} x_{m_2}^{(l_2)} Y_{m_2}^{(l_2)}(\theta,\psi)) Y_m^{(l)}(\theta,\psi) \sin\theta \mathrm{d}\theta \mathrm{d}\psi$$

(57)

Let $\mathbf{x}^{*(l_1)} = w_{l_1} \mathbf{x}^{(l_1)}$ and $\mathbf{x}^{*(l_2)} = w_{l_2} \mathbf{x}^{(l_2)}$, then Eqn. (57) actually calculates the coefficients of function $F_3^*(\theta,\psi) = F_1^*(\theta,\psi) \cdot F_2^*(\theta,\psi)$, where $F_1^*(\theta,\psi) = \sum_{l_1=0}^\infty \sum_{m_1=-l_1}^{l_1} x_{m_1}^{*(l_1)} Y_{m_1}^{(l_1)}(\theta,\psi)$ and $F_2^*(\theta,\psi) = \sum_{l_2=0}^\infty \sum_{m_2=-l_2}^{l_2} x_{m_2}^{*(l_2)} Y_{m_2}^{(l_2)}(\theta,\psi)$. To this end, we can use the efficient approach presented in Section 3.2 to accelerate such computation. After we calculate the coefficients of $F_3^*(\theta,\psi)$, we then multiply the weight $w_l$ with corresponding irreps. Thus, the setting of weighted combinations are naturally included in our Gaunt Tensor Product.

**Equivariant Convolutions.** One special case of Equivariant Feature Interaction is to instantiate $\tilde{\mathbf{y}}^{(L_2)}$ as spherical harmonics filters, i.e., $\sum_{l_1=0}^{L_1} \sum_{l_2=0}^{L_2} h_{l_1,l_2}^l (\mathbf{x}_i^{(l_1)} \otimes_{cg} Y^{(l_2)}(\frac{\mathbf{r}_i - \mathbf{r}_j}{||\mathbf{r}_i - \mathbf{r}_j||}))^{(l)}$, where $\mathbf{r}_i, \mathbf{r}_j \in \mathbb{R}^3$ are positions of objects, $Y^{(l_2)} : S^2 \rightarrow \mathbb{R}^{2l_2+1}$ is the type-$l_2$ spherical harmonic, $h_{l_1,l_2}^l \in \mathbb{R}$ are learnable weights calculated based on the relative distance and types of objects $i, j$, e.g., $h_{l_1,l_2}^l = F_{l,l_1,l_2}(||\mathbf{r}_i - \mathbf{r}_j||, x_i, x_j)$, where $F_{l,l_1,l_2}$ are instantiated as Gaussian basis functions and $x_i, x_j$ are the type of object $i, j$. Similarly, $h_{l_1,l_2}^l$ can be reparameterized for our approach.

Moreover, Passaro & Zitnick (2023) provides an interesting observation that if we select a rotation $g \in SO(3)$ with $\mathbf{D}^{(1)}(g) \frac{\mathbf{r}_i - \mathbf{r}_j}{||\mathbf{r}_i - \mathbf{r}_j||} = (0, 1, 0)$, then $Y_m^{(l)}(\mathbf{D}^{(1)}(g) \frac{\mathbf{r}_i - \mathbf{r}_j}{||\mathbf{r}_i - \mathbf{r}_j||}) \propto \delta_m^{(l)}$, i.e., being non-zero only if $m = 0$. Such a sparsity further propagates to its coefficients of the 2D Fourier basis, which brings further acceleration in Eqn. (6): Let $y_m^{(l)}$ denote the elements of the spherical harmonic filter. Due to the sparsity of $y_m^{(l)} = \delta_m^{(l)}$ and coefficients $y_{u,v}^{l,m,*}$, the conversion rule from spherical harmonics to 2D Fourier bases can be rewritten as:

$$y_{u,0}^* = \sum_{l=0}^L y_0^{(l)} y_{u,0}^{l,0,*}, \quad -L \le u \le L,$$
$$y_{u,v}^* = 0, \quad \text{if } v \ne 0.$$

(58)

The coefficients of the spherical harmonic filters in the 2D Fourier basis are also sparse, and such a conversion has an $\mathcal{O}(L^2)$ complexity instead.

**Equivariant Many-body Interactions.** Operations in this class perform multiple tensor products among equivariant features, i.e., $\tilde{\mathbf{x}}_1^{(L_1)} \otimes_{\tilde{c}g} \tilde{\mathbf{x}}_2^{(L_2)} \otimes_{\tilde{c}g} ... \otimes_{\tilde{c}g} \tilde{\mathbf{x}}_n^{(L_n)}$ with $n-1$ tensor products. For exam-

ple, given equivariant features $A_i^{(L')} \in \mathbb{R}^{(L'+1)^2}$ of atom $i$, MACE (Batatia et al., 2022b) constructs the many-body features by $\sum_{\eta_\nu} \sum_{\mathbf{lm}} C_{\eta_\nu,\mathbf{lm}}^{LM} \prod_{\xi=1}^\nu A_{i,m_\xi}^{(l_\xi)}$, where $A_{i,m_\xi}^{(l_\xi)}$ is the $m_\xi$-th element of the type-$l_\xi$ irrep of $A_i^{(L')}$, $\mathbf{lm}$ are the multi-indices $(l_1 m_1, ..., l_\nu m_\nu)$, $C_{\eta_\nu,\mathbf{lm}}^{LM}$ are the generalized Clebsch-Gordan coefficients, i.e., $C_{l_1 m_1, ..., l_n m_n}^{LM} = C_{l_1 m_1, l_2 m_2}^{L_2 M_2} C_{L_2 M_2, l_3 m_3}^{L_3 M_3} ... C_{L_{N-1} M_{N-1}, l_N m_N}^{L_N M_N}$ with $L = (L_2, ..., L_N), |l_1 - l_2| \leq L_2 \leq l_1 + l_2, \forall i \geq 3 |L_{i-1} - l_i| \leq L_i \leq L_{i-1} + l_i$ and $M_i \in \{m_i| - l_i \leq m_i \leq l_i\}$, $\eta_\nu$ enumerates all possible couplings of $l_1, ..., l_\nu$. In particular, it is the explicit form of performing tensor products with $A_i^{(L)}$ itself $\nu$ times via the generalized Clebsch-Gordan coefficients:

$$B_{\nu,i} = A_i^{(L')} \otimes_{\tilde{cg}} ... \otimes_{\tilde{cg}} A_i^{(L')}. \tag{59}$$

We generalize the Gaunt coefficients to this setting. Let $G_{(l_1,m_1)...(l_n,m_n)}^{(L,M)} = \int_0^{2\pi} \int_0^\pi Y_{m_1}^{(l_1)}(\theta,\psi) Y_{m_2}^{(l_2)}(\theta,\psi)...Y_{m_n}^{(l_n)}(\theta,\psi) Y_M^{(L)}(\theta,\psi) \sin\theta \mathrm{d}\theta \mathrm{d}\psi$. Then Eqn. (59) in the generalized Gaunt coefficients is equivalent to:

$$(A_i^{(L')} \underbrace{\otimes_{\tilde{Gaunt}} \cdots \otimes_{\tilde{Gaunt}}}_{n-1 \text{ times}} A_i^{(L')})_m^{(l)}$$

$$= \int_0^{2\pi} \int_0^\pi \left( \sum_{l_1=0}^{L'} \sum_{m_1=-l_1}^{l_1} A_{i,m_1}^{(l_1)} Y_{m_1}^{(l_1)}(\theta,\psi) \right) ... \left( \sum_{l_n=0}^{L'} \sum_{m_n=-l_n}^{l_n} A_{i,m_n}^{(l_n)} Y_{m_n}^{(l_n)}(\theta,\psi) \right) Y_m^{(l)}(\theta,\psi) \sin\theta \mathrm{d}\theta \mathrm{d}\psi. \tag{60}$$

In fact, we actually compute multiple multiplications among multiple spherical functions. We can also use the parameterization tricks to similarly multiply each $A_i^{(l)}$ with learnable weights $w_l$ to achieve weighted $(\mathbf{lm}) \to l$ combinations. The approach presented in Section 3.2 can be naturally used: after the conversion from spherical harmonics to 2D Fourier bases, we use $\mathbf{A}_i^{\nu*}, \nu = 1, ..., n$ to denote the coefficients of 2D Fourier bases of these $n$ spherical functions, and the 2D Convolution can be compactly expressed as:

$$\mathbf{A}_i^{1*} \circledast \mathbf{A}_i^{2*} \circledast ... \circledast \mathbf{A}_i^{(n-1)*} \circledast \mathbf{A}_i^{n*}, \tag{61}$$

where $\circledast$ denotes the convolution. Interestingly, these 2D Convolutions have associativity, and we can use a divide-and-conquer approach for parallelizing the computation of these convolution operations. For example, if we perform the tensor product operation among 8 functions, the coefficients $\mathbf{A}_i^{\nu*}$ can be divided into pair-wise groups for parallel computation, i.e., $(1,2) \to 1*, (3,4) \to 2*, (5,6) \to 3*, (7,8) \to 4*$. Such a procedure is iteratively conducted until the final results are calculated. The computational complexity of performing 2D convolutions in the Equivariant Many-body Interaction setting is $\mathcal{O}(n^2 L^2 \log L)$, while recursively computing Eqn. (61) has an $\mathcal{O}(n^3 L^2 \log L)$ complexity.

**Discussions.** In practice, it is common in models to maintain $C$ channels of equivariant features, i.e., $\tilde{\mathbf{X}}_i^{(L)} \in \mathbb{R}^{(L+1)^2 \times C}$ denotes the equivariant features of object $i$, which contains $C$ channels of irreps $\tilde{\mathbf{x}}_{i,c}^{(L)}, 0 \leq c < C$ of degrees up to $L$. Our approach can be naturally extended by simply defining the combination rules of features in different channels:

- Channel-wise: given $\tilde{\mathbf{X}}_i^{(L)}, \tilde{\mathbf{X}}_j^{(L)} \in \mathbb{R}^{(L+1)^2 \times C}$, output features in the $c$-th channel are calculated based on $\tilde{\mathbf{x}}_{i,c}^{(L)}$ and $\tilde{\mathbf{x}}_{j,c}^{(L)}$ only ($\mathcal{O}(C)$ full tensor products):

$$(\tilde{\mathbf{X}}_i^{(L)} \otimes_{\tilde{Gaunt}}^{\mathbf{W}_{\text{wise}}} \tilde{\mathbf{X}}_j^{(L)})_c = \tilde{\mathbf{x}}_{i,c}^{(L)} \otimes_{\tilde{Gaunt}}^{\mathbf{w}_c} \tilde{\mathbf{x}}_{j,c}^{(L)}. \tag{62}$$

- Channel-mixing: given $\tilde{\mathbf{X}}_i^{(L)}, \tilde{\mathbf{X}}_j^{(L)} \in \mathbb{R}^{(L+1)^2 \times C}$, output features in the $c$-th channel are calculated based on weighted combinations of $\tilde{\mathbf{x}}_{i,c_1}^{(L)}, c_1 = 0, 1, ..., C$ and $\tilde{\mathbf{x}}_{j,c_2}^{(L)}, c_2 = 0, 1, ..., C$ ($\mathcal{O}(C^2)$ full tensor products):

$$(\tilde{\mathbf{X}}_i^{(L)} \otimes_{\tilde{Gaunt}}^{\mathbf{W}_{\text{wise}}} \tilde{\mathbf{X}}_j^{(L)})_c = \sum_{c_1=0}^C \sum_{c_2=0}^C w_{c_1,c_2}^c \cdot (\tilde{\mathbf{x}}_{i,c_1}^{(L)} \otimes_{\tilde{Gaunt}}^{\mathbf{w}} \tilde{\mathbf{x}}_{j,c_2}^{(L)}). \tag{63}$$

Our approach is also not limited to full tensor products only. Given $\mathbf{x}^{(l_1)} \otimes_{Gaunt} \mathbf{x}^{(l_2)}$ only, the approach presented in Section 3.2 can still be used:

$$
\begin{aligned}
(\mathbf{x}^{(l_1)} \otimes_{Gaunt} \mathbf{x}^{(l_2)})_m^{(l)} &= \sum_{m_1=-l_1}^{l_1} \sum_{m_2=-l_2}^{l_2} G_{(l_1,m_1)(l_2,m_2)}^{(l,m)} x_{m_1}^{(l_1)} x_{m_2}^{(l_2)}, \\
&= \sum_{m_1=-l_1}^{l_1} \sum_{m_2=-l_2}^{l_2} x_{m_1}^{(l_1)} x_{m_2}^{(l_2)} \int_0^{2\pi} \int_0^{\pi} Y_{m_1}^{(l_1)}(\theta,\psi) Y_{m_2}^{(l_2)}(\theta,\psi) Y_m^{(l)}(\theta,\psi) \sin\theta \mathrm{d}\theta \mathrm{d}\psi, \\
&= \int_0^{2\pi} \int_0^{\pi} \left( \sum_{l_1=0}^{L_1} \sum_{m_1=-l_1}^{l_1} x_{m_1}^{(l_1)} Y_{m_1}^{(l_1)}(\theta,\psi) \right) \left( \sum_{l_2=0}^{L_2} \sum_{m_2=-l_2}^{l_2} x_{m_2}^{(l_2)} Y_{m_2}^{(l_2)}(\theta,\psi) \right) Y_m^{(l)}(\theta,\psi) \sin\theta \mathrm{d}\theta \mathrm{d}\psi.
\end{aligned}
\tag{64}
$$

where we set $x_{m_u}^{(l_u)}, x_{m_v}^{(l_v)}$ to 0 for $l_u \neq l_1, l_v \neq l_2$. Our Gaunt Tensor Product between $\mathbf{x}^{(l_1)}$ and $\mathbf{x}^{(l_2)}$ can still be equivalently viewed as a multiplication between two spherical functions, which can be efficiently calculated by using the approach presented in Section 3.2.

## D    PROOF OF THE EQUIVARIANCE OF THE GAUNT TENSOR PRODUCT

In this section, we provide formal proof of the equivariance of our Gaunt Tensor Product. As stated in Section 2, tensor products of irreps via the Clebsch-Gordan coefficients are equivariant to the $O(3)$ group. Given $\mathbf{x}^{(l_1)}$ and $\mathbf{y}^{(l_2)}$, we have the following equation for any $g \in O(3)$:

$$
\mathbf{D}^{(l)}(g)(\mathbf{x}^{(l_1)} \otimes_{cg} \mathbf{y}^{(l_2)})^{(l)} = \left( (\mathbf{D}^{(l_1)}(g)\mathbf{x}^{(l_1)}) \otimes_{cg} (\mathbf{D}^{(l_2)}(g)\mathbf{x}^{(l_2)}) \right)^{(l)}.
\tag{65}
$$

Let $\mathbf{D}_{mn}^{(l)}(g)$ denote the $(m,n)$-th element of the $\mathbf{D}^{(l)}(g)$ matrix. First, the $m$-th element of $\mathbf{D}^{(l)}(g)(\mathbf{x}^{(l_1)} \otimes_{cg} \mathbf{y}^{(l_2)})^{(l)}$ can be explicitly written via the Clebsch-Gordan coefficients:

$$
\begin{aligned}
\mathbf{D}^{(l)}(g)(\mathbf{x}^{(l_1)} \otimes_{cg} \mathbf{y}^{(l_2)})_m^{(l)} &= \sum_{m'} \mathbf{D}_{mm'}^{(l)}(g)(\mathbf{x}^{(l_1)} \otimes_{cg} \mathbf{y}^{(l_2)})_{m'}^{(l)} \\
&= \sum_{m'} \mathbf{D}_{mm'}^{(l)}(g) \left( \sum_{m_1,m_2} C_{(l_1,m_1),(l_2,m_2)}^{(l,m')} x_{m_1}^{(l_1)} y_{m_2}^{(l_2)} \right) \\
&= \sum_{m_1,m_2} \left( \sum_{m'} \mathbf{D}_{mm'}^{(l)}(g) C_{(l_1,m_1),(l_2,m_2)}^{(l,m')} \right) x_{m_1}^{(l_1)} y_{m_2}^{(l_2)}.
\end{aligned}
\tag{66}
$$

Second, the $m$-th element of $((\mathbf{D}^{(l_1)}(g)\mathbf{x}^{(l_1)}) \otimes_{cg} (\mathbf{D}^{(l_2)}(g)\mathbf{x}^{(l_2)}))^{(l)}$ can also be explicitly written via the Clebsch-Gordan coefficients:

$$
\begin{aligned}
\left( (\mathbf{D}^{(l_1)}(g)\mathbf{x}^{(l_1)}) \otimes_{cg} (\mathbf{D}^{(l_2)}(g)\mathbf{x}^{(l_2)}) \right)_m^{(l)} &= \sum_{m_1,m_2} C_{(l_1,m_1),(l_2,m_2)}^{(l,m)} \left( \sum_{m_1'} \mathbf{D}_{m_1 m_1'}^{(l_1)}(g) x_{m_1'}^{(l_1)} \right) \left( \sum_{m_2'} \mathbf{D}_{m_2 m_2'}^{(l_2)}(g) y_{m_2'}^{(l_2)} \right) \\
&= \sum_{m_1,m_2} \sum_{m_1',m_2'} C_{(l_1,m_1),(l_2,m_2)}^{(l,m)} \mathbf{D}_{m_1 m_1'}^{(l_1)}(g) \mathbf{D}_{m_2 m_2'}^{(l_2)}(g) x_{m_1'}^{(l_1)} y_{m_2'}^{(l_2)} \\
&= \sum_{m_1,m_2} \left( \sum_{m_1',m_2'} C_{(l_1,m_1'),(l_2,m_2')}^{(l,m)} \mathbf{D}_{m_1' m_1}^{(l_1)}(g) \mathbf{D}_{m_2' m_2}^{(l_2)}(g) \right) x_{m_1}^{(l_1)} y_{m_2}^{(l_2)}.
\end{aligned}
\tag{67}
$$

We can conclude that the Clebsch-Gordan coefficients have the following properties:

$$
\sum_{m'} \mathbf{D}_{mm'}^{(l)}(g) C_{(l_1,m_1),(l_2,m_2)}^{(l,m')} = \sum_{m_1',m_2'} C_{(l_1,m_1'),(l_2,m_2')}^{(l,m)} \mathbf{D}_{m_1' m_1}^{(l_1)}(g) \mathbf{D}_{m_2' m_2}^{(l_2)}(g).
\tag{68}
$$

The Gaunt coefficients also satisfy these constraints. Here, we use the Cartesian coordinates as the input of spherical harmonics and denote the Gaunt coefficients $G_{(l_1,m_1),(l_2,m_2)}^{(l,m)}$ as

$\int_{S^2} Y_{m_1}^{(l_1)}(\tilde{\mathbf{r}})Y_{m_2}^{(l_2)}(\tilde{\mathbf{r}})Y_m^{(l)}(\tilde{\mathbf{r}})\mathrm{d}\tilde{\mathbf{r}}$. We have the following equations:

$$
\begin{aligned}
\sum_{m'}\mathbf{D}_{mm'}^{(l)}(g)G_{(l_1,m_1),(l_2,m_2)}^{(l,m')} &= \sum_{m'}\mathbf{D}_{mm'}^{(l)}(g)\int_{S^2} Y_{m_1}^{(l_1)}(\tilde{\mathbf{r}})Y_{m_2}^{(l_2)}(\tilde{\mathbf{r}})Y_m^{(l)}(\tilde{\mathbf{r}})\mathrm{d}\tilde{\mathbf{r}} \\
&= \int_{S^2} Y_{m_1}^{(l_1)}(\tilde{\mathbf{r}})Y_{m_2}^{(l_2)}(\tilde{\mathbf{r}})\left(\sum_{m'}\mathbf{D}_{mm'}^{(l)}(g)Y_m^{(l)}(\tilde{\mathbf{r}})\right)\mathrm{d}\tilde{\mathbf{r}} \qquad (69) \\
&= \int_{S^2} Y_{m_1}^{(l_1)}(\tilde{\mathbf{r}})Y_{m_2}^{(l_2)}(\tilde{\mathbf{r}})Y_m^{(l)}(\mathbf{D}^{(1)}(g)\tilde{\mathbf{r}})\mathrm{d}\tilde{\mathbf{r}}.
\end{aligned}
$$

$$
\begin{aligned}
&\sum_{m_1',m_2'} G_{(l_1,m_1'),(l_2,m_2')}^{(l,m)}\mathbf{D}_{m_1'm_1}^{(l_1)}(g)\mathbf{D}_{m_2'm_2}^{(l_2)}(g) \\
&= \sum_{m_1',m_2'}\mathbf{D}_{m_1'm_1}^{(l_1)}(g)\mathbf{D}_{m_2'm_2}^{(l_2)}(g)\int_{S^2} Y_{m_1'}^{(l_1)}(\tilde{\mathbf{r}})Y_{m_2'}^{(l_2)}(\tilde{\mathbf{r}})Y_m^{(l)}(\tilde{\mathbf{r}})\mathrm{d}\tilde{\mathbf{r}} \\
&= \int_{S^2}\left(\sum_{m_1'}\mathbf{D}_{m_1'm_1}^{(l_1)}(g)Y_{m_1'}^{(l_1)}(\tilde{\mathbf{r}})\right)\left(\sum_{m_2'}\mathbf{D}_{m_2'm_2}^{(l_2)}(g)Y_{m_2'}^{(l_2)}(\tilde{\mathbf{r}})\right)Y_m^{(l)}(\tilde{\mathbf{r}})\mathrm{d}\tilde{\mathbf{r}} \\
&= \int_{S^2}\left(\sum_{m_1'}\mathbf{D}_{m_1m_1'}^{(l_1)}(g)^\top Y_{m_1'}^{(l_1)}(\tilde{\mathbf{r}})\right)\left(\sum_{m_2'}\mathbf{D}_{m_2m_2'}^{(l_2)}(g)^\top Y_{m_2'}^{(l_2)}(\tilde{\mathbf{r}})\right)Y_m^{(l)}(\tilde{\mathbf{r}})\mathrm{d}\tilde{\mathbf{r}} \qquad (70) \\
&= \int_{S^2}\left(\sum_{m_1'}\mathbf{D}_{m_1m_1'}^{(l_1)}(g)^{-1} Y_{m_1'}^{(l_1)}(\tilde{\mathbf{r}})\right)\left(\sum_{m_2'}\mathbf{D}_{m_2m_2'}^{(l_2)}(g)^{-1} Y_{m_2'}^{(l_2)}(\tilde{\mathbf{r}})\right)Y_m^{(l)}(\tilde{\mathbf{r}})\mathrm{d}\tilde{\mathbf{r}} \\
&= \int_{S^2}\left(\sum_{m_1'}\mathbf{D}_{m_1m_1'}^{(l_1)}(g^{-1}) Y_{m_1'}^{(l_1)}(\tilde{\mathbf{r}})\right)\left(\sum_{m_2'}\mathbf{D}_{m_2m_2'}^{(l_2)}(g^{-1}) Y_{m_2'}^{(l_2)}(\tilde{\mathbf{r}})\right)Y_m^{(l)}(\tilde{\mathbf{r}})\mathrm{d}\tilde{\mathbf{r}} \\
&= \int_{S^2} Y_{m_1}^{(l_1)}(\mathbf{D}^{(1)}(g^{-1})\tilde{\mathbf{r}})Y_{m_2}^{(l_2)}(\mathbf{D}^{(1)}(g^{-1})\tilde{\mathbf{r}})Y_m^{(l)}(\tilde{\mathbf{r}})\mathrm{d}\tilde{\mathbf{r}}.
\end{aligned}
$$

Due to the symmetry of the unit sphere $S^2$, we have $\int_{S^2} Y_{m_1}^{(l_1)}(\tilde{\mathbf{r}})Y_{m_2}^{(l_2)}(\tilde{\mathbf{r}})Y_m^{(l)}(\mathbf{D}^{(1)}(g)\tilde{\mathbf{r}})\mathrm{d}\tilde{\mathbf{r}} = \int_{S^2} Y_{m_1}^{(l_1)}(\mathbf{D}^{(1)}(g^{-1})\tilde{\mathbf{r}})Y_{m_2}^{(l_2)}(\mathbf{D}^{(1)}(g^{-1})\tilde{\mathbf{r}})Y_m^{(l)}(\tilde{\mathbf{r}})\mathrm{d}\tilde{\mathbf{r}}$. Therefore, the Gaunt coefficients also satisfy the properties to make our Gaunt Tensor Product equivariant to the $O(3)$ group.

# E    EXPERIMENTAL DETAILS

## E.1    EFFICIENCY COMPARISONS

**Settings.**    (1) For the Equivariant Feature Interaction operation, we use the e3nn implementation (Geiger & Smidt, 2022) as the baseline. We randomly sample 10 pairs of features containing irreps of up to degree $L$ with 128 channels. We evaluate different degrees of $L$ and report the average inference time[4]. (2) For the Equivariant Convolution operation, we use the eSCN implementation (Passaro & Zitnick, 2023) as the baseline, due to its enhanced efficiency over e3nn. In this setting, 10 pairs of features and spherical harmonics filters are sampled instead. The number of channels is kept as 128. We also evaluate different degrees of $L$. We implement our Gaunt Tensor Product by using the insights of eSCN for further acceleration, as stated in Section 3.3. (3) For the Equivariant Many-body Interaction operation, we compare our approach with both the e3nn and the MACE implementations (Batatia et al., 2022b). We evaluate different degrees of $L$ and the number of tensor product operands $\nu$: (a) fix $\nu = 3$ and vary $L$, (b) fix $L = 2$ and vary $\nu$. The benchmarking results are obtained by using the same NVIDIA Tesla V100 GPU.

## E.2    SANITY CHECK

**Baselines.**    In this experiment, we choose the SEGNN (Brandstetter et al., 2022) as the baseline, which achieves state-of-the-art performance on the N-body simulation task. In particular, SEGNN proposes a generalization of equivariant GNNs such that node and edge attributes are not restricted to scalars. Based on tensor products of irreps, SEGNN injects geometric and physical quantities into node updates. To investigate the underlying effects of different parameterizations between our Gaunt Tensor Product and the Clebsch-Gordan Tensor Product, we compare two architectures: (1) SEGNN with Clebsch-Gordan Tensor Product implementation; (2) SEGNN with Gaunt Tensor Product. For other configurations, we follow Brandstetter et al. (2022) to keep the same.

**Settings.**    Following Brandstetter et al. (2022), we use the N-body simulation task to investigate the effects of different parameterizations. The N-body particle system consists of 5 particles that carry a positive or negative charge, having initial position and velocity in a 3-dimensional space. The task is to estimate all particle positions after 1,000 timesteps. We use the 4-layer SEGNN and set the max degree of attributes ($l_a$) irreps in [1, 2]. The max degree of feature vectors ($l_f$) is set to 1. The learning rate is set to 0.01 for $l_a = 1$ and 0.02 for $l_a = 2$. The learning rate was reduced using an on-plateau scheduler based on the validation loss with patience of 10 and a decay factor of 0.1 for $l_a = 1$ and 0.2 for $l_a = 2$. Other settings are kept the same as the original model. The model is trained on 1 NVIDIA Tesla V100 GPU.

## E.3    OC20 S2EF

**Datasets.**    OC20 is a diverse and large-scale dataset consisting of 1.2M DFT relaxation trajectories, which are computed by using the revised Perdew-Burke-Ernzerhof (RPBE) functional (Hammer et al., 1999). Each structure in OC20 has an adsorbate molecule placed on a catalyst surface, and the core task is Structure-to-Energy-Forces (S2EF), which is to predict the energy and force mean absolute error (MAE). The "All" split of OC20 contains 134M training examples. Due to limited computational resources, we adopt the widely used S2EF-2M subset, which consists of 2 million training examples in total. Four splits of validation sets are used including in-distribution (ID), out-of-distribution on adsorbates (OOD-Ads), out-of-distribution on catalysts (OOD-Cat), and out-of-distribution on both adsorbates and catalysts (OOD-Both). The evaluation metrics include the Energy MAE and Force MAE, the cosine similarity of predicted forces and ground-truth forces (Force Cos), and the ratio of examples on which the predicted energy and forces are under a predefined threshold (EFwT).

**Baselines.**    We compare our approach based on the EquiformerV2 (Liao et al., 2024) with several competitive baselines from the leaderboard of the Open Catalyst Challenge (Chanussot et al., 2021). First, we compare with several classic invariant neural networks. SchNet (Schütt et al., 2018) leveraged the interatomic distances encoded via radial basis functions, which serve as the weights of

---

[4]We noticed that after running the same e3nn code multiple times, the subsequent executions are faster than the initial attempt, because of the code compilation. Nevertheless, our approach remains more efficient overall.

continuous-filter convolutional layers. DimeNet++(Gasteiger et al., 2020a) built upon the DimeNet model (Gasteiger et al., 2020b) which introduced the directional message passing that encodes both distance and angular information between triplets of atoms. SpinConv (Shuaibi et al., 2021) used a per-edge local coordinate frame and a spin convolution over the remaining degree of freedom to build the model. GemNet (Gasteiger et al., 2021) embedded all atom pairs within a given cutoff distance based on interatomic directions, and proposed three forms of interaction to update the directional embeddings: Two-hop geometric message passing (Q-MP), one-hop geometric message passing (T-MP), and atom self-interactions. An efficient variant named GemNet-T is proposed to use cheaper forms of interaction. Based on GemNet, Gasteiger et al. (2022) comprehensively investigated the relationships between the performance and chemical diversity, system size, dataset size and domain shift. From these results, an efficient improvement to GemNet architectures was developed, named GemNet-OC.

Recently, SCN (Zitnick et al., 2022) used a rotation trick to encode the atomic environment by using the irreps, while not using the tensor product operations. Note that SCN does not strictly obey the equivariance constraint. eSCN (Passaro & Zitnick, 2023) presented an interesting observation that the spherical harmonics filters have specific sparsity patterns if a proper rotation acts on the input spherical coordinates. Combined with patterns of the Clebsch-Gordan coefficients, eSCN proposed an efficient equivariant convolution to reduce $SO(3)$ tensor products to $SO(2)$ linear operations, achieving efficient computation. Built upon the eSCN convolution, EquiformerV2 (Liao et al., 2024) developed a scalable Transformer-based model, which achieved state-of-the-art performance on the OC20 and AdsorbML (Lan et al., 2022) benchmarks.

**Architectures.** EquiformerV2 is composed of stacked equivariant Transformer blocks. Each block consists of two layers: an Equivariant Graph Attention layer followed by a feed-forward layer, with both layers having separable equivariant Layer Normalization and skip connections. EquiformerV2 can use irreps of higher degrees, e.g., 4 or 6, with the eSCN convolution. However, such implementations also restrict the available equivariant operations for this scalable architecture. In this work, we proposed the Gaunt Tensor Product, which serves as a new method for efficient equivariant operations, enabling further combinations between architectures like EquiformerV2 and effective equivariant operations. As an example, we use our Gaunt Tensor Product to implement the Selfmix layer in the Equivariant Feature Interaction operation class. This layer receives equivariant features of each atom as input. For each atom, full tensor products of its irreps are performed with weighted combinations. In each EquiformerV2 block, we insert this layer with the equivariant Layer Normalization in between the Equivariant Graph Attention layer and the feed-forward layer. For this modified architecture, the original implementation would be extremely slow because to the limitation of the eSCN implementation. In contrast, our Gaunt Tensor Product has high efficiency when using irreps of higher degrees.

**Settings.** Following Liao et al. (2024), we use a 12-layer EquiformerV2 model. The max degree of irreps is set in [4, 6]. The dimension of hidden layers and feed-forward layers is set to 128. The number of attention heads is set to 8. The number of radial Basis kernels is set to 600. We use AdamW as the optimizer (Kingma & Ba, 2014). The gradient clip norm is set to 100.0. The peak learning rate is set to 2e-4. The batch size is set to 64. The dropout ratios is set to 0.1. The ratio of stochastic depth (Huang et al., 2016) is set to 0.001. The weight decay is set to 0.0. The model is trained for 12 epochs with a 0.1-epoch warm-up stage. After the warm-up stage, the learning rate decays by using a cosine learning rate scheduler. The batch size is set to 64. Other hyperparameters are kept the same as EquiformerV2 for a fair comparison. The model is trained on 16 NVIDIA Tesla V100 GPUs.

## E.4    3BPA

**Datasets.**    Following Batatia et al. (2022b), we use the 3BPA dataset to benchmark the Equivariant Many-body Interaction operations. The 3BPA dataset contains geometric structures of a flexible drug-like organic molecule 3-(benzyloxy)pyridin-2-amine sampled from different temperature molecular dynamics trajectories. The training set of the 3BPA dataset contains 500 geometries sampled at the 300 K temperature, while the three test sets contain geometries sampled at 300K, 600 K, and 1200 K to assess in- and out-of-domain accuracy. A fourth test set includes optimized geometries, in which two dihedral angles of the molecule are fixed, while a third angle is varied between 0 and 360 degrees. This variation produces *dihedral slices* that traverse regions of the potential energy surface (PES), different from the training data. This test assesses the smoothness and accuracy of the potential energy surface (PES) region that dictates the conformers present in a simulation. Consequently, it directly influences properties of interest, such as the binding free energies associated with protein targets.

**Baselines.** We follow (Batatia et al., 2022b; Musaelian et al., 2023) to choose several competitive baselines for comparison. Atomic Cluster Expansion (Drautz, 2019) provided a systematic framework for constructing high body order complete polynomial basis functions, which includes many existing atomic environment representations as special cases (Behler & Parrinello, 2007; Bartók et al., 2010; Thompson et al., 2015; Shapeev, 2016). sGDML (Chmiela et al., 2019) is an optimized implementation of the symmetric gradient domain machine learning model, which is able to faithfully reproduce global potential energy surfaces (PES) for molecules with a few dozen atoms from a limited number of user-provided reference molecular conformations and the associated atomic forces. NequIP (Batzner et al., 2022) extended the tensor product operations to be $E(3)$-equivariant. Allegro (Musaelian et al., 2023) further developed a strictly local equivariant network to improve scalability. Batatia et al. (2022a) provided an analysis on the design space of $E(3)$-equivariant operations and developed BoTNet. MACE (Batatia et al., 2022b) combined the message passing neural networks with Equivariant Many-body Interaction operations, which achieved better data efficiency and improved generalization performance.

**Settings.** Based on the MACE architecture (Batatia et al., 2022b), we compare our Gaunt Tensor Product with the implementations of e3nn and MACE. Following Batatia et al. (2022b), we use a two-layer MACE model. The dimension of hidden layers is set to 256. Radial features are generated using 8 Bessel basis functions and a polynomial envelope for the cutoff with p = 5. The radial features are fed to an MLP of size [64, 64, 64, 1024], using SiLU nonlinearities on the outputs of the hidden layers. The readout function of the first layer is implemented as a simple linear transformation. The readout function of the second layer is a single-layer MLP with 16 hidden dimensions. We use the AMSGrad variant of Adam as the optimizer, and set the hyper-parameter $\epsilon$ to 1e-8 and $(\beta_1, \beta_2)$ to (0.9,0.999). The batch size is set to 5. The peak learning rate is set to 1e-2. The learning rate was reduced using an on-plateau scheduler based on the validation loss with patience of 50 and a decay factor of 0.8. We use an exponential moving average with a weight of 0.99 to evaluate the validation set as well as for the final model. We set the exponential weight decay to 5e-7. The model is trained on 1 NVIDIA A6000 GPU.

