# OpenReview forum: "Enabling Efficient Equivariant Operations in the Fourier Basis via Gaunt Tensor Products"
_ICLR.cc/2024/Conference — ICLR 2024 spotlight_

### Official Review · Reviewer_1Msw · 2023-10-27

**Soundness:** 3 good
**Presentation:** 3 good
**Contribution:** 3 good
**Rating:** 8
**Confidence:** 3

**Summary:**

This paper introduces the Gaunt Tensor Product, an efficient method to compute the tensor products of (high-degree) irreps as compared to the widely-used Clebsch-Gordan (CG) product. This is established through the connection between CG coefficients and Gaunt coefficients via the Wigner-Eckart theorem, and this connection implies that tensor product can be equivalently represented as multiplication of spherical functions. Through convolution theorem and FFT, the computational cost has been reduced from $O(L^6)$ to $O(L^3)$, where $L$ is the highest-degree of the irreps.

**Strengths:**

1. The paper is well-structured and articulate. The authors have included most of the requisite background material to the appendix, thereby maintaining the flow of the main text.

2. The paper's approach to enhancing computational efficiency, specifically through establishing a connection between CG coefficients, Gaunt coefficients, and the FFT, is both innovative and compelling.

3. The experimental evaluations comparing the proposed tensor product with the conventionally employed CG product are both extensive and convincing.

**Weaknesses:**

1. One point of critique is related to the treatment of Theorem 3.1 in the manuscript. The text uses a range of specialized terms—such as "spherical tensor operator," "reduced matrix element," and "total angular momentum"—and introduces various notations that may not be readily accessible to readers unfamiliar with quantum mechanics. To enhance clarity and comprehension, I recommend that the authors elaborate on these concepts, particularly in their exposition of the Wigner-Eckart theorem.

2. Another inquiry concerns the necessity of using high-degree irreps. In most E3-NN architectures, irreps with degrees no greater than L=2 are commonly employed. Consequently, I question whether the computational overhead incurred through the FFT in implementing the proposed tensor product is actually more costly than directly using the CG product for low-degree irreps.

**Questions:**

Please refer to the weakness section.

---

> ### Author Response · Authors · 2023-11-21
> **Response to Reviewer 1Msw (Part 1/2)**
>
> Thank you for supporting our work. We reply to all of your questions below, and we have carefully revised and updated our paper to include these new changes:
>
> > **Elaborate on concepts in Theorem 3.1**
>
> Thank you for this suggestion. We have revised our paper to thoroughly elaborate on the mentioned notations and concepts in Section 3.1, for improved clarity and readability. In detail, we added a new subsection (Appendix A.7, Page 19-26), which is composed of the following contents:
>
> - Appendix A.7 Angular Momentum, Tensor Operators and Wigner-Eckart theorem
>   - A.7.1 Preliminary
>     - Hilbert space and Dirac notation ("bra-ket" notation)
>     - Operators
>     - Eigenkets, basis kets and matrix elements of operators
>   - A.7.2 A Brief Overview of Angular Momentum in Quantum Mechanics
>     - Momentum Operator and Position Operator
>     - Angular Momentum Operator
>     - Eigenstates and eigenvalues of Angular Momentum Operators
>     - Addition of Angular Momentum Operators
>   - A.7.3 Tensor Operators and Wigner-Eckart theorem
>     - Rotation Operators
>     - Vector Operators and Tensor Operators
>     - Wigner-Eckart theorem
>
> For completeness, we comprehensively introduce necessary mathematical concepts in quantum mechanics, and elaborate step-by-step on their exposition in the context of Section 3.1 in an easy-to-follow and self-consistent manner, especially for general audiences who are unfamiliar with quantum mechanics. In Section 3.1, we add a reference to Appendix A.7 for interested readers, in case they are curious about the details of the notations and concepts. This should allow our readers to have more context about the concepts in Section 3.1, and also provide information on the relationship between Clebsch-Gordan coefficients and Gaunt coefficients.

---

> ### Author Response · Authors · 2023-11-21
> **Response to Reviewer 1Msw (Part 2/2)**
>
> > **Necessity of using higher-degree irreps**
>
> We would like to clarify the following points and the motivation for using irreps of higher degrees:
>
> 1) The reason that most E3NN architectures commonly use irreps with degrees no greater than $L=2$ lies in the *significant computational cost*, rather than it being unnecessary to use. In the literature of E3NN architectures, the $\mathcal{O}(L^6)$ computational complexity of computing the Tensor Products of irreps has been widely acknowledged as a major obstacle. In Section 1 of our paper, we refer the readers to see comments in more than 10 related works over almost three years, discussing such limitations, which indeed reflects *the difficulty of tackling this problem*. Due to this, most existing E3NN architectures *are substantially limited* and cannot use irreps with higher degrees in practice, not because there is no need.
>
> 2) Using irreps of higher degrees is indeed *necessary*. Even though the computational costs limit the usage of irreps of higher degrees, there are multiple examples showing that higher degree irreps can improve performance. Recent advancements in E(3)-equivariant networks suggest that the empirical performance of state-of-the-art models based on the Tensor Products of irreps can be consistently boosted by using higher degrees of irreps [1,2]. Additionally, specific tasks require the model to use higher degrees of irreps. For example, the DFT Hamiltonian prediction task [3,4] requires the model to predict a molecular system's DFT Hamiltonian matrix, whose shape is determined by the number of orbitals used to describe the system (proportional to the number of electrons). To construct the predicted matrix, E(3)-equivariant networks based on irreps are utilized, and the shape of Hamiltonian matrices exactly determines the largest degree of irreps that are necessary to use. Therefore, it is necessary, and has great significance, to tackle this bottleneck efficiency issue so that we can appropriately use irreps of higher degrees.
>
> 3) Our approach still remains very efficient when we use lower degree irreps. We provide detailed statistics of the three settings from Figure 1:
>
>   - Equivariant Feature Interaction: 7.9x and 17.4x times acceleration for $L=2$ and $L=3$ respectively (compared to the e3nn implementation)
>   - Equivariant Convolution: 4.1x and 7.0x times acceleration for $L=2$ and $L=3$ respectively (compared to the eSCN implementation)
>   - Equivariant Many-body Interaction: 193.6x and 288.7x times acceleration for $L=2$ and $L=3$ respectively (compared to the e3nn implementation)
>
>
> [1] Passaro, S., & Zitnick, C. L. (2023). Reducing SO (3) Convolutions to SO (2) for Efficient Equivariant GNNs. *ICML 2023*.
>
> [2] Liao, Y. L., Wood, B., Das, A., & Smidt, T. (2023). EquiformerV2: Improved Equivariant Transformer for Scaling to Higher-Degree Representations. *arXiv preprint arXiv:2306.12059*.
>
> [3] Unke, O., Bogojeski, M., Gastegger, M., Geiger, M., Smidt, T., & Müller, K. R. (2021). SE (3)-equivariant prediction of molecular wavefunctions and electronic densities. *Advances in Neural Information Processing Systems*, *34*, 14434-14447.
>
> [4] Yu, H., Xu, Z., Qian, X., Qian, X., & Ji, S. (2023). Efficient and Equivariant Graph Networks for Predicting Quantum Hamiltonian. *ICML 2023*.
>
> We thank you again for your efforts in reviewing our paper, and we have carefully revised our paper and replied to each of your comments. We look forward to your re-evaluation of our submission based on our responses and updated results.

---

> > ### Comment · Reviewer_1Msw · 2023-11-22
> >
> > I thank the authors for the detailed response. I am raising my rating.

---

> ### Author Response · Authors · 2023-11-22
> **Thank you for the prompt feedback**
>
> Thank you for the re-evaluation of our work! We enjoyed the discussion with you, and thank you for helping us improve the quality of our paper.

---

### Official Review · Reviewer_AAYb · 2023-10-30

**Soundness:** 4 excellent
**Presentation:** 4 excellent
**Contribution:** 3 good
**Rating:** 6
**Confidence:** 3

**Summary:**

This work introduces a new framework for calculating the equivariant operations efficiently. The key idea is to convert the Clebasch-Gordan coefficients to Gaunt coefficients, by which the tensor product of irreps can be calculated efficiently in the Fourier domain.

**Strengths:**

1. The paper is well-written, and the main idea is explained clearly.
2. It is smart and interesting that the basis transformation is applied to the Fourier domain. The usefulness of the proposed method for improving computational efficiency is verified with numerical experiments.

**Weaknesses:**

It would be better if more discussion could be given about the added value of this work. It will help the readers who are not familiar with this direction (like me) to easily understand the contribution and main differences compared with the existing works.

**Questions:**

1. Could you give me more explanation about the *braket* notations used in Theorem 3.1? It seems not clearly introduced in the paper.
2. Could you explain more or give the intuition about the irreducible representation?
3. How to determine in practice the truncation, ie. the degree L, in the 2D Fourier expansion (above Eq. (5))?
4. Is it possible to choose other basis systems instead of Fourier? May I know how choosing different systems affects the result of this paper?

---

> ### Author Response · Authors · 2023-11-21
> **Response to Reviewer AAYb (Part 1/7)**
>
> Thank you for your comments, and we appreciate your feedback. We are glad to see that you found our work to be smart and interesting. We respond to all of your questions below.
>
> > **More discussion on the added value of this work.**
>
> We elaborate on the significance of this work, and our contributions, for a more general audience:
>
> - **The significance of Tensor Products of Irreducible Representations**: As we discuss in the introduction, it is well-known that numerous fundamental real-world problems exhibit symmetries to the Euclidean group. In recent years, we have witnessed numerous advances in the scientific domain, including molecular modeling (see ACM Gordon Bell Prize attributed to machine learning potentials [1,2]), protein biology (AlphaFold2 [Jumper et al., 2021]), and 3D vision. A key component for the success of these breakthroughs was through enforcing the equivariant constraints to the Euclidean group on neural networks. This is because the objects of interest in these applications naturally reside in the 3D Euclidean space.
>
>   To enable the enforcement of this equivariance, we use irreducible representations (irreps) as the smallest and most complete representation of the $O(3)$ group (the group of all rotations and inversions on Euclidean space). Specifically, the key computation to enforce equivariance involves the Tensor Products of irreducible representations. This has played a key role in pushing the frontier of E(3)-equivariant operations with both theoretical completeness (universal approximation capability [Dym & Maron, 2021]) and state-of-the-art performance on real-world applications [Ramakrishnan et al., 2014; Chmiela et al., 2017; Chanussot et al., 2021].
>   *Therefore, there is great significance in investigating the limitations and challenges of Tensor Product of irreducible representations, with the aim of more efficient and effective E(3)-equivariant operations for real-world applications*.
>
> - **The limitations and challenges of Tensor Products of irreducible representations:** Despite the theoretical completeness and empirical superiority of using these tensor product operations, the computational complexity inherent to computing the Tensor Products of irreps is very expensive, and is known to be a hurdle in achieving greater efficiency (and thus effectiveness) in performance:
>
>   - *Limited Efficiency*: Empirically, it has been seen that higher symmetries (represented by $L$) tend to have better accuracy.
>   As analyzed in our paper, the full tensor product of irreps up to degree $L$ has an *$\mathcal{O}(L^6)$ complexity*. Therefore, equivariant operations based on Tensor products of Irreps are usually the computational bottleneck of E(3)-equivariant networks. This means that using such operations on large 3D systems is severely impeded.
>   - *Restricted Effectiveness*: due to the $\mathcal{O}(L^6)$ complexity, most existing works have to limit $L$ to 2 or 3 in practice. However, recent advancements suggest that the empirical performance of state-of-the-art models based on such operations can be consistently boosted by using irreps of higher degrees. Therefore, the computational complexity of computing the Tensor Products of irreducible representations also hinders the advancement of more effective and powerful E(3)-equivariant networks.
>
>   In fact, such limitations have been widely recognized as a critical challenge in the literature of E(3)-equivariant operations. In the introduction section of our paper, we refer the readers to see comments in more than 10 related works in almost three years discussing such limitations, which reflects *both the great significance and difficulty in tackling this problem*.

---

> ### Author Response · Authors · 2023-11-21
> **Response to Reviewer AAYb (Part 2/7)**
>
> - **Our contributions**: In this paper, we introduce a systematic methodology to significantly accelerate the computation of the Tensor Products of irreps. Our main contributions can be summarized into the following points:
>
>   - **Original and deep insights**: We are the first to provide a new perspective on the computation of the Tensor Products of irreps. In section 3.1, we first reveal the underlying connection between Clebsch-Gordan coefficients and Gaunt coefficients via the Wigner-Eckart theorem. This connection motivates the derivation of a new understanding: the tensor product of irreps via the Gaunt coefficients is equivalent to multiplication between spherical functions represented by linear combinations of spherical harmonic bases. Our insights present added value to the broad community on equivariant networks in multiple ways:
>
>     - *Accelerating the computation of the Tensor Products of irreps*: The difficulty in tackling this efficiency issue is mainly due to the lack of understanding of the underlying mechanism. Our perspective naturally motivates a systematic framework to significantly accelerate the computation process.
>     - *Providing new opportunities for a deeper understanding of equivariant operations*: Beyond motivating efficiency, our method also provides a new perspective to the community. This new perspective may have potential impacts on future advancements to create even more effective and powerful E(3)-equivariant networks.
>
>   - **A systematic framework for efficient equivariant operations**: We propose a systematic framework (Section 3.2) to significantly accelerate the computation of the Tensor Products of irreps. Our efficient computational methodology using the convolution theorem and FFTs also presents added value to the broad community:
>
>     - *Efficiency*: The computational complexity of the Tensor Products of irreps is substantially reduced from $\mathcal{O}(L^6)$ to $\mathcal{O}(L^3)$. As verified in the experimental section, our approach achieves significant acceleration and memory reduction for advanced E(3)-equivariant models on popular benchmarks. The achieved acceleration significantly reduces the costs of training and inference, allowing our method to be applied to large 3D systems.
>     - *Effectiveness*: Our approach enables the usage of irreps with higher degrees. As shown in Figure 1, the computational costs consistently keep low when we increase the degrees of irreps, while using irreps of higher degrees can consistently boost empirical performance and accuracy on challenging tasks.
>
>   - **General applicability**: As computing the Tensor Products of irreps are key to E(3)-equivariant operations, researchers have developed several approaches to compute these operations in the past few years. In fact, another added value to the community, brought by our approach, lies in its *general applicability*.
>
>     In Section 3.3, we provide a comprehensive study on representative operation classes that are widely used (see the number of related works in each class) in advanced equivariant models. Note that developing a unified approach that can accelerate all these operation classes is indeed challenging, especially for the Equivariant Many-Body Interaction operations. We find out that our new perspective on Clebsch-Gordan coefficients and Gaunt coefficients can be generalized to the many-body setting, and correspondingly we develop a generalized acceleration methodology in this case. *We are able to unify and significantly accelerate different kinds of E(3)-equivariant operations that are based on the Tensor Product of irreps*.
>
>   *Combining the above contributions, our work moves the field forward towards more efficient and effective E(3)-equivariant operations for real-world applications.*

---

> ### Author Response · Authors · 2023-11-21
> **Response to Reviewer AAYb (Part 3/7)**
>
> - **Comparisons with previous approaches**: As stated in the introduction and Appendix B, many previous approaches developed E(3)-equivariant operations based on the Tensor Product of irreps. However, though the limitations and challenges of computing these operations have already been widely acknowledged in the past few years, only a few works have looked at this problem in detail, because of the difficulty:
>
>   - ICML23, eSCN [3]: In this work, the authors proposed an efficient method for one special class of E(3)-equivariant operations based on the Tensor Products of irreps: Equivariant Convolution (as introduced in Section 3.3 of our paper). In this special case, the authors introduced sparsity to spherical harmonic filters via a rotation transformation on the input, which provided opportunities for acceleration.
>   - NeurIPS22, MACE [4]: In this work, the authors proposed a new machine learning force field model based on Equivariant Many-Body Interaction operations. Though the focus of the paper is not on the acceleration of tensor products, the authors provided an efficient approach to compute its operation by trading memory for time (thus, we see the model has out-of-memory issues, in Table 2).
>
>   Our work has a number of differences from these previous works:
>
>   - Our work provides a systematic and unified framework for multiple equivariant operation classes. As previously mentioned, none of the previous works are able to construct a unified framework to accelerate different kinds of E(3)-equivariant operations based on the Tensor Products of irreps, which also reflects the difficulty of this critical challenge.
>   - Our work provides new insights on the underlying mechanism of the core of these operations, which has further significance on understanding and advancing E(3)-equivariant operations in multiple ways.
>   - Compared to eSCN's/MACE's approach on the corresponding special operation classes, our approach can still achieve significant acceleration and substantially reduced memory costs, as shown in Figure 1 and Table 2 in our paper.
>
> [1] https://www.acm.org/media-center/2020/november/gordon-bell-prize-2020
>
> [2] https://seas.harvard.edu/news/2023/10/kozinsky-and-team-among-finalists-gordon-bell-prize
>
> [3] Passaro, S., & Zitnick, C. L. (2023). Reducing SO (3) Convolutions to SO (2) for Efficient Equivariant GNNs. *ICML 2023*.
>
> [4] Batatia, I., Kovacs, D. P., Simm, G., Ortner, C., & Csányi, G. (2022). MACE: Higher order equivariant message passing neural networks for fast and accurate force fields. *Advances in Neural Information Processing Systems*, *35*, 11423-11436.

---

> ### Author Response · Authors · 2023-11-21
> **Response to Reviewer AAYb (Part 4/7)**
>
> > **Provide more explanations on the bracket notations used in Theorem 3.1**
>
> We have updated the paper to provide a comprehensive overview in Appendix A.7 on Dirac notation (bra-ket notation) and additional quantum mechanics concepts. We provide additional explanation here and summarize what we added:
>
> In physics, bra-ket notation is used to describe objects in a Hilbert space $\mathcal{H}$. It is a convenient way to write out inner product operations $(\cdot,\cdot):\mathcal{H}\times\mathcal{H}\to\mathbb{C}$. Vectors in the Hilbert space $\mathcal{H}$ are written as kets: $|\alpha\rangle\in\mathcal{H}$, and each ket has the one-to-one corresponding bra denoted as $\langle\alpha|$, with which the inner product of two kets can be written as $(|\alpha\rangle,|\beta\rangle)\equiv\langle\alpha|\beta\rangle$.
>
> To describe the mapping between kets in a Hilbert space, we use the concept of operators, i.e., $\hat{O}:\mathcal{H}\to\mathcal{H}$. We define the eigenkets (also called eigenstates) $|\alpha\rangle$ of an operator as $\hat{A}$ if $\hat{A}|\alpha\rangle=a|\alpha$, where the scalar $a$ is the corresponding eigenvalue. Each ket in a Hilbert space can also be expressed as a linear combination of the basis kets. For any operator $\hat{A}$ and fixed basis kets $|\alpha_i\rangle$, we can calculate its matrix elements via $\langle\alpha_i|\hat{A}|\alpha_j\rangle$.
>
> Building on the above notation, we provide a detailed explanation on the bra-ket notation used in Theorem 3.1:
>
>   - $|l,m\rangle$ denote the eigenstates of angular momentum operators.
>
>     The angular momentum operators involve two operators: the momentum operator $\hat{\mathbf{p}}=-i\hbar\left(\frac{\partial}{\partial x},\frac{\partial}{\partial y},\frac{\partial}{\partial z}\right)$ and the position operator $\hat{\mathbf{x}}=\left(\hat{x},\hat{y},\hat{z}\right)$, where the position operator $\hat{x}$ (also $\hat{y},\hat{z}$) satisfies $\hat{x}f(x)\equiv xf(x)$. The angular momentum operator is defined as $\hat{\mathbf{L}}=(\hat{L}_x,\hat{L}_y,\hat{L}_z)$, where $\hat{L}_x=\hat{y} \hat{p}_z-\hat{z} \hat{p}_y,\hat{L}_y=\hat{z} \hat{p}_x-\hat{x} \hat{p}_z,\hat{L}_z=\hat{x} \hat{p}_y-\hat{y} \hat{p}_x$. The square of $\hat{\mathbf{L}}$ is $\hat{\mathbf{L}}^2=\hat{L}_x^2+\hat{L}_y^2+\hat{L}_z^2$. The kets $|l,m\rangle$ are simultaneously the solutions of the following equations (1) $\hat{L}_z|l,m\rangle=\hbar m|l,m\rangle$; (2) $\hat{\mathbf{L}}^2|l,m\rangle=\hbar l(l+1)|l,m\rangle$. Thus, $|l,m\rangle$ denote the eigenstates of the angular momentum operators. Given $l\in\mathbb{N}$, we have $-l\leq m\leq l$, and the set of $|l,m\rangle$ forms a basis for the $2l+1$-dimensional Hilbert space.
>
>     The explicit form of kets $|l,m\rangle$ is exactly the spherical harmonics, $Y^{(l)}_m$, introduced in Section 2 in our paper. In quantum mechanics, it is conventional to use the ket notation $|l,m\rangle$, and we can obtain the function value of $Y^{(l)}_m$ with input $(\theta,\phi)$ via $\langle\theta,\phi|l,m\rangle\equiv Y^{(l)}_m(\theta,\phi)$.
>
>   - $\langle l,m|$ denotes the bra corresponding to the eigenstates $|l,m\rangle$.
>
>   - $\langle J,M|T^{(j)}_m|l,n\rangle$ denotes the matrix elements of spherical tensor operator $T^{(j)}_m$. This is when the basis corresponds to eigenkets of the angular momentum operators. As previously introduced, by fixing $J,l$, there are $2J+1$ and $2l+1$ basis kets (specified as $|J,M\rangle$ and $|l,n\rangle$) for the $2J+1$-dimensional and $2l+1$-dimensional Hilbert spaces respectively. By definition, spherical tensor operators of rank $j$ are families $\mathbf{T}^{(j)}=(T^{(j)}\_{-j},...,T^{(j)}_j)^\top$ of $2j+1$ operators $T^{j}_m$ that transform by rotations, like spherical harmonics. Therefore, there are $(2J+1)\times(2j+1)\times(2l+1)$ matrix elements of the spherical tensor operator $\mathbf{T}^{(j)}$ under the $|J,M\rangle$ and $|l,n\rangle$ basis.
>
>   - $\langle J || \mathbf{T}^{(j)}||l\rangle$ denotes the reduced matrix elements, which are constants that only depend on the value of $J,j,l$.

---

> ### Author Response · Authors · 2023-11-21
> **Response to Reviewer AAYb (Part 5/7)**
>
> We added a new subsection (Appendix A.7, Page 19-26) that includes more detail, and is composed of the following contents:
>
> - Appendix A.7 Angular Momentum, Tensor Operators and Wigner-Eckart theorem
>   - A.7.1 Preliminary
>     - Hilbert space and Dirac notation ("bra-ket" notation)
>     - Operators
>     - Eigenkets, basis kets and matrix elements of operators
>   - A.7.2 A Brief Overview of Angular Momentum in Quantum Mechanics
>     - Momentum Operator and Position Operator
>     - Angular Momentum Operator
>     - Eigenstates and eigenvalues of Angular Momentum Operators
>     - Addition of Angular Momentum Operators
>   - A.7.3 Tensor Operators and Wigner-Eckart theorem
>     - Rotation Operators
>     - Vector Operators and Tensor Operators
>     - Wigner-Eckart theorem
>
> For completeness, we comprehensively introduce the necessary mathematical concepts in quantum mechanics, and elaborate step-by-step on their exposition in the context of Section 3.1. We hope that this should help general audiences who are unfamiliar with quantum mechanics. In Section 3.1, we add a reference to Appendix A.7 for interested readers, in case they are curious about the details of the notations and concepts. This should allow our readers to have more context about the concepts in Section 3.1, and capture the underlying mechanisms of the relationship between Clebsch-Gordan coefficients and Gaunt coefficients.

---

> ### Author Response · Authors · 2023-11-21
> **Response to Reviewer AAYb (Part 6/7)**
>
> > **More explanation on irreducible representations**
>
> We provide more information on irreducible representations below, and include examples:
>
> In Section 2 of our paper, we briefly introduced the concept of group and group representations. We know that the concept of groups is useful to describe symmetries and transformations in objects, which has an abstract algebra structure revealing the interactions between group elements. To instantiate the abstract elements into specific actions on vector spaces, we use group representations.
>
> As an example, we look at the $SO(3)$ rotation group. Now consider two sets. The first set consists of all 3D rotations, while the second set consists of all three-dimensional orthogonal transformation matrices with the determinant being 1. As we already know, both sets satisfy the definition of group, i.e., the associated interaction operations (successive rotations and matrix multiplications) of these two sets satisfy group axioms (introduced in Appendix A.1). For example, rotating a rod about its center axis by 90 degrees clockwise, followed by another rotation by 90 degrees clockwise, forms a rotation of 180 degrees clockwise. Matrix multiplication between two orthogonal transformation matrices still results in an orthogonal transformation matrix. Moreover, each rotation corresponds to an orthogonal transformation matrix, i.e., the orthogonal transformation matrices are exactly the representations of the set of rotations on the three-dimensional vector space.
>
> Note that the representations of the rotation group are not limited only to the three-dimensional vector space. For any vector space $V$, each $g\in SO(3)$ has its corresponding matrix representation in this space, which act on vectors in $V$ via matrix-vector multiplications. Let us consider a system of two electrons located in the three-dimensional vector space with Cartesian coordinates $\vec{r}_1=(x_1,y_1,z_1)$ and $\vec{r}_2=(x_2,y_2,z_2)$. If we apply the same rotation $g\in SO(3)$ to both electrons, we obtain the new coordinates of these two electrons $\vec{r}_1'=(x_1',y_1',z_1')$ and $\vec{r}_2'=(x_2',y_2',z_2')$. By using $\mathbf{R}_g\in\mathbb{R}^{3\times 3}$ to denote the representation of $g$ in $\mathbb{R}^3$, we have $\vec{r}_1'=\mathbb{R}_g\vec{r}_1$ and $\vec{r}_2'=\mathbb{R}_g\vec{r}_2$. If we consider the position of the whole system, we actually obtain $\vec{r}=(x_1,y_1,z_1,x_2,y_2,z_2)$ and $\vec{r}'=(x_1',y_1',z_1',x_2',y_2',z_2')$, which is in the six-dimensional vector space. Hence, we obtain the representation of $g$ on this six-dimensional space as a matrix $\mathbf{R}_g'$: $\left(\begin{array}{cc}
> \mathbf{R}_g & 0 \newline
> 0 & \mathbf{R}_g
> \end{array}\right)$ in the shape of $6\times 6$, and $\vec{r}'=\mathbf{R}_g'\vec{r}$. This representation is actually composed of two representations on the three-dimensional space, which is the subspace of the six-dimensional space. This phenomenon says that the representation of $g$ on the six-dimensional space is *reducible* and can be reduced into two subrepresentations on the subspace. The key idea from these examples is that irreducible representations cannot be further reduced into representations on the subspace.
> As stated in Section 2, we introduce the concept of Wigner-D matrix, which is exactly the irreducible representation of $SO(3)$ on spaces with $2l+1$ dimension, $l\in\mathbb{N}$. Any matrix representation of $SO(3)$ can thus be reduced to an equivalent block diagonal matrix representation, with Wigner-D matrices along the diagonal.
>
> > **How to determine, in practice, the truncation in the 2D Fourier expansion**
>
> In practice, the truncation of the 2D Fourier expansion, i.e., the degree $L$, is set to the same value as the maximum degree of irreps.

---

> ### Author Response · Authors · 2023-11-21
> **Response to Reviewer AAYb (Part 7/7)**
>
> > **Choosing other basis systems, and possible effects**
>
> This is a good point. As introduced in Section 3, our efficient approach derives from the new perspective on the Tensor Products of irreps via Gaunt coefficients: the tensor product of irreps via the Gaunt coefficients is equivalent to multiplication between spherical functions represented by linear combinations of spherical harmonic bases. Given this, it is natural to use 2D Fourier basis to represent spherical functions via a change of basis, because we can leverage efficient computational methods like FFTs for acceleration. In fact, there also exist other 2D bases to represent spherical functions, e.g, 2D Haar Wavelets, which are possible to use instead of a Fourier basis. In our work, we provide the following criteria for choosing the different basis:
> - *Approximation quality*: the multiplication results between spherical functions should not differ when using different bases, i.e., the approximation error of the change of basis should be small enough. In our work, we provided empirical evidence on the satisfactory approximation quality of the 2D Fourier basis in the sanity check experiment with SEGNN on the N-body simulation task. In Figure 1, we can see that our Gaunt Tensor Product achieves comparable, and even better performance compared to the standard implementation, which verifies the approximation quality of using a 2D Fourier basis.
> - *Computational complexity*: another critical point to consider is the computational complexity of calculating the multiplication between spherical functions after the change of basis. In Section 3.2 of our paper, we provide a thorough analysis on the computational complexity by choosing the 2D Fourier basis, which requires the following steps to be efficient enough to achieve acceleration:
>   - Converting coefficients from spherical harmonics to a new basis.
>   - Perform multiplication between spherical functions in the new basis.
>   - Converting coefficients from new basis back to spherical harmonics.
>
>   Since the computational complexity of the standard implementation of CG tensor products is $\mathcal{O}(L^6)$, the above three steps should thus have lower complexity than this.
>
> We thank you again for your efforts in reviewing our paper, and we have carefully revised our paper and replied to each of your comments. We look forward to your re-evaluation of our submission based on our responses and updated results.

---

> ### Comment · Reviewer_AAYb · 2023-11-23
>
> Thank you for the detailed reply. My questions have been well answered.

---

> > ### Author Response · Authors · 2023-11-23
> > **Thank you for the feedback**
> >
> > Thank you for your response, and we're glad to see that all your questions have been well answered. We enjoyed having the discussion. Given this, we'd appreciate it if you could re-evaluate our work and update your score in light of our updates.

---

### Official Review · Reviewer_wTba · 2023-11-01

**Soundness:** 4 excellent
**Presentation:** 3 good
**Contribution:** 3 good
**Rating:** 8
**Confidence:** 4

**Summary:**

The authors propose a systematic approach to accelerate the computation of tensor products of irreps by connecting Clebsch-Gordan coefficients to Gaunt coefficients. Based on this, the authors introduce a change of basis from spherical harmonics to a 2D Fourier basis. This transformation enables efficient computation via the convolution theorem and Fast Fourier Transforms (FFT), reducing the complexity from $O(L^6)$ to $O(L^3)$.

**Strengths:**

- The paper is very well-structured, featuring clear and logical derivations. The authors deserve commendation for their efforts in showing the equivariance of tensor products with Gaunt coefficients (Appendix D).
- Interpreting the spherical tensor product as a multiplication of spherical functions and subsequently transforming it into a 2D convolution provides valuable insight into the underlying operations. In my opinion, this perspective is more important than the subsequent application of FFT, which is a logical extension.
- The substantial reduction in time and memory cost on the 3BPA dataset serves as compelling evidence of the method's effectiveness.

**Weaknesses:**

Majors:
- The current derivation and application are closely tied to $\text{O}(3)$. Is it feasible to extend this convolutional perspective beyond $\text{O}(3)$, e.g., to other Lie groups like $\text{SU}(2)$? It would be beneficial to see discussions on the aspect of its generalization.
- I would like to see the inference time of EquiformerV2 model with and without the proposed Gaunt tensor products.

Minors:
- Please adjust Equation (23)(26)(36) as they are written in the margins of the paper.

**Questions:**

See weaknesses.

---

> ### Author Response · Authors · 2023-11-21
> **Response to Reviewer wTba**
>
> Thank you for your recognition of the significance and the potential impact of our work. We also appreciate your comments to further improve our work. Here are our responses to your questions:
>
> > **Extending our work to other groups like SU(2)**
>
> This is a good point. In the literature, other Lie groups also describe important symmetries exhibited in real-world applications, e.g., $SU(2^N)$ in quantum mechanics, $SU(3)$ in quantum chromodynamics, and $SO^{+}(1,3)$. Equivariant models respecting these symmetries are demonstrated to be powerful in these tasks [1, 2]. It is thus interesting to extend our methodology to these scenarios. In practice, there are two points we need to check:
> - The framework to respect symmetries: The equivariant networks in our paper respect symmetries based on the Tensor Products of irreps. To extend our approach to other Lie groups, we need to also generalize the Tensor Products of Irreps for these groups. This includes (1) constructing irreps of the target Lie group; (2) decomposing the Tensor Products of irreps by using generalized Clebsch-Gordan coefficients of the targeted group; (3) computing the tensor product results.
> - Relationship between Clebsch-Gordan coefficients and Gaunt coefficients: For other Lie groups, it is crucial to check the relationship between the generalized version of Clebsch-Gordan coefficients to Gaunt coefficients, which determines whether we can express the computation of tensor products as multiplications between functions expanded by bases.
>
> In our preliminary study, we have several interesting results that correspond to the above two points. First, we noticed that there exists a recent work [3] that proposed a general framework to respect symmetries to Reductive Lie Groups including ($SO^+(1,3)$,$SU(N)$ and etc.), which generalizes the Tensor Product of irreps and demonstrates the feasibility to other groups. The presented framework exactly matches our first point, thus meaning that it could be possible to apply our approach here. Moreover, to check the relationship between the Clebsch-Gordan coefficients and Gaunt coefficients, we use the Wigner-Eckart theorem. In fact, we can generalize the Wigner-Eckart theorem presented in Section 3.1 of our paper to other Lie groups, which is also supported in [4]. The provided generalized Wigner-Eckart theorem serves as a powerful tool to examine the relationships between different groups. From these preliminary results, we believe it is feasible to further explore the possibility of our methodology for other Lie groups, and leave this for future work.
>
>
> > **Inference time comparison between EquiformerV2 with and without the proposed Gaunt Tensor Product operations**
>
> We provide inference time comparisons here. In our experiments, we use the EquiformerV2 as the base architecture on the OC20 S2EF task to demonstrate the effectiveness of our approach. We apply our Gaunt Tensor Product to construct an efficient Equivariant Feature Interaction operation called Selfmix, and add it to each layer of the EquiformerV2 model. As we introduced in our paper, the equivariant operations in EquiformerV2 cannot be used to accelerate the computation of operations in the Equivariant Feature Interaction class. In contrast, our Gaunt Tensor Product can provide significant acceleration here. Empirically, this operation in EquiformerV2 with the Gaunt Tensor Product is 5.14x times faster than that without Gaunt Tensor Product on the OC20 S2EF task, demonstrating the efficiency of our approach.
>
> > **Equations in margins of the paper**
>
> Thank you for pointing this out. We have fixed this in our revised paper.
>
> [1] Favoni, M., Ipp, A., Müller, D. I., & Schuh, D. (2022). Lattice gauge equivariant convolutional neural networks. Physical Review Letters, 128(3), 032003.
>
> [2] Bogatskiy, A., Anderson, B., Offermann, J., Roussi, M., Miller, D., & Kondor, R. (2020, November). Lorentz group equivariant neural network for particle physics. In International Conference on Machine Learning (pp. 992-1002). PMLR.
>
> [3] Batatia, I., Geiger, M., Munoz, J., Smidt, T., Silberman, L., & Ortner, C. (2023). A General Framework for Equivariant Neural Networks on Reductive Lie Groups. arXiv preprint arXiv:2306.00091.
>
> [4] Lang, L., & Weiler, M. (2020). A Wigner-Eckart Theorem for Group Equivariant Convolution Kernels. ICLR 2021.
>
> We thank you again for your efforts in reviewing our paper, and we have carefully revised our paper and replied to each of your comments. It is interesting to study the questions you raised, and we believe these discussions are valuable to further explore the potential impact of our work.

---

### Public Comment · ~Mit_Kotak1 · 2023-11-23
**Note about JAX Implementation and Clebsch-Gordan Tensor Products**

# Note about e3nn JAX Implementation and Clebsch-Gordan Tensor Products

We greatly enjoyed your paper on the Gaunt Tensor Products! We believe it can unlock many different applications for equivariant neural networks.

While reading your paper, we had two observations that may be of interest:
* We would like to point out that e3nn has a JAX implementation of the Clebsch-Gordan Tensor Product (CGTP): https://e3nn-jax.readthedocs.io/en/latest/. With the SphericalSignal API in e3nn-jax, we were also able to make an implementation of the Gaunt Tensor Product (GTP) in JAX. We benchmarked these JAX and PyTorch e3nn implementations of the CGTP and the GTP below. We find that while the GTP is significantly more efficient in PyTorch, it seems that the CGTP is more efficient in JAX upto L = 6 on a RTX A5500 GPU and L = 4 on a T4 GPU. We have uploaded our benchmarking code here: https://gist.github.com/mitkotak/fcc0564fce02a8410fbcc73bfbaeb449. We would appreciate it if you could share your implementation to compare as well.

* Another difference between the CGTP and the GTP that was not discussed in your manuscript is that the GTP cannot compute certain features due to symmetry constraints. For example, the CGTP between two vectors contains a psuedovector (corresponding to the cross-product of these two vectors). This term however, does not show up in the GTP because the GTP satisfies GTP(x, y) = GTP(y, x) for any vectors x, y. This makes the pseudovector term cancel out. While hard to account for such cancellation in the benchmarking, it would be great if you can mention the fact that the GTP missses some terms found in the CGTP.

Sorry for bringing this up at the very end of the discussion period; we saw this paper very recently and wanted to verify our benchmarking claims before posting. We believe that this paper represents a very important contribution for building equivariant neural networks.

-- Ameya Daigavane and Mit Kotak

---

### Meta-Review · Area_Chair_WW7d · 2023-12-07

**Metareview:**

An important component of $O(3)$-equivariant models defined using irreducible representations (irreps) is taking tensor products of irreps. The current paper suggests a new perspective that allows an efficient computation of these products as an alternative to the Clebsch-Gordan formula. The main idea is to relate tensor product to products of spherical functions using a tool called Gaunt coefficients, and second, using FFT for fast computation of the respective tensor products. There is a large computation complexity saving in the form of reducing $O(L^6)$ to $O(L^3)$ in complexity when using irreps up to order $L$. Reviewers find this approach to be compelling both in its core idea and its practical implications. The authors also made an honest effort in making their paper accessible to a wider audience in their revision.

**Justification For Why Not Higher Score:**

The full practical implications, e.g., the benefit in high order tensor product in equivariant models is still not clear.

**Justification For Why Not Lower Score:**

The paper presents a novel idea improving an important aspect of a rather common equivariant architecture.

---

### Decision · Program_Chairs · 2024-01-16

Accept (spotlight)